# NLI: N̲on-uniform L̲inear I̲nterpolation Approximation of Nonlinear Operations for Efficient LLMs Inference

**Jiangyong Yu**[1*]          **Xiaomeng Han**[2*]          **Xing Hu**[1]

**Chen Xu**[1]          **Zhe Jiang**[2]          **Dawei Yang**[1✉]

[1]**Houmo AI**          [2]**Southeast University**

## ABSTRACT

Large Language Models (LLMs) have demonstrated remarkable performance across a wide range of tasks, but their deployment is often constrained by substantial memory footprints and computational costs. While prior work has achieved significant progress in compressing and accelerating linear layers, nonlinear layers—such as SiLU, RMSNorm, and Softmax—still heavily depend on high-precision floating-point operations. In this paper, we propose a calibration-free, dynamic-programming-optimal, and hardware-friendly framework called N̲on-uniform L̲inear I̲nterpolation (NLI). NLI is capable of efficiently approximating a variety of nonlinear functions, enabling seamless integration into LLMs and other deep neural networks with almost no loss in accuracy. NLI ingeniously recasts cutpoint selection as a dynamic-programming problem, achieving the *globally* minimal interpolation error in $\mathcal{O}(M \times N^2)$ time via Bellman's optimality principle. Based on the NLI algorithm, we also design and implement a plug-and-play universal nonlinear computation unit. Hardware experiments demonstrate that the NLI Engine achieves more than 4× improvement in computational efficiency compared to the state-of-the-art designs.

## 1 INTRODUCTION

Large language models (LLMs; e.g., GPT3-175B Zong & Krishnamachari (2022), LLaMA3-405B Touvron et al. (2023), and Deepseek-R1-671B Guo et al. (2024)) have achieved remarkable success across various domains, such as text translation Tekgurler (2025), image classification Naeem et al. (2023) and text generation Li et al. (2024). However, due to the massive model size, LLMs impose significant demands on both memory bandwidth and computation. This makes edge deployment of LLMs extremely challenging, hindering the further application of LLMs.

Recent research efforts have focused on using low-bit, high-efficiency data formats Lin et al. (2024) to improve the computational efficiency of linear layers. For example, SmoothQuant Xiao et al. (2023) achieves W8A8 integer quantization by smoothing activation values. OSTquant Hu et al. (2025) optimizes the distribution of weights and activations through orthogonal transformations and scaling, enabling W4A8 integer quantization. Moreover, many hardware architectures support low-bit linear computations, further enhancing the efficiency of linear layers. For example, the Tensor Cores in the NVIDIA H100 NVIDIA (2023) natively support INT8 linear operations. Gemmini Genc et al. (2021) also supports INT8 linear operations. However, the computation of nonlinear layers (e.g., Softmax, RMSNorm, SiLU) in LLMs still heavily depends on high-precision floating-point formats (such as FP32), resulting in substantial computational overhead. This further exacerbates the performance disparity between linear and nonlinear operations in LLM inference. For example, in the H100 SXM5, the FP16 linear computational power is 1024 × greater than that of the special function units,

---

[*]Equal contribution
[✉]Corresponding author

a) Range of SiLU activations across Large Language Models

b) Performance Comparison of Non-linear Approximation Methods in Large Language Models

Figure 1: **(a)** Range of SiLU activations in representative LLMs; values can exceed $\pm 100$ (e.g., Qwen2.5-32B, Qwen3-8B). **(b)** Wikitext-2 perplexity (log-scale) with FP32, NN-LUT, and our **NLI**. NN-LUT collapses when outliers occur—perplexity skyrockets up to $7.0 \times 10^4$—whereas NLI matches FP32 across scales.

yet in scenarios with a head attention size of 128, the demand for linear computational power is 256 $\times$ that of nonlinear computational power.

Nonlinear functions in LLMs typically involve hardware-intensive transcendental functions (e.g., exp) and some complicated algebraic functions (e.g., square root, and reciprocal functions). Some prior works accelerate these functions via hardware-friendly approximations. For example, Softermax Stevens et al. (2021) uses low-precision arithmetic to implement `Softmax` operations through a base replacement strategy. I-Bert Kim et al. (2021) proposed approximation techniques to compute `GELU`, `Softmax`, and `LayerNorm` using INT32 arithmetic. Although these designs achieve high efficiency and accuracy for certain nonlinear operations in the BERT model, they do not reliably transfer to other LLMs with hundreds of billions of parameters (e.g., LLaMA, Qwen, OPT). Moreover, their inherent hardware inflexibility further limits their applicability in NPUs. Other researchers have attempted to propose more general-purpose methods. For example, NN-LUT Yu et al. (2022) computes nonlinear functions using first-order derivative fitting ($y = sx + t$), offering good generality within a certain input range. To maintain model accuracy, it introduces a data calibration method. Although NN-LUT works well for the modest activation spans found in early models such as BERT, its coverage collapses when confronted with the extreme outliers characteristic of LLMs. As shown in Figure 1(a), the `SiLU` inputs of seven representative LLMs frequently exceed $\pm 100$. In contrast, NN-LUT was designed and validated only for the input domain $(-5, 5)$, so these inputs clearly lie beyond its expected scope. When the same NN-LUT configuration is applied to these out-of-range values, the approximation error compounds throughout the network and the model fails to converge, producing a Wikitext-2 perplexity surge of up to $7 \times 10^4$ (Figure 1 (b)). Taken together, for LLMs at tens to hundreds of billions of parameters, current acceleration strategies for nonlinear functions remain insufficient. This combination of narrow validity ranges, reliance on calibration, and hardware inflexibility further limits robustness and deployability.

In this paper, we propose a calibration-free, dynamic-programming-optimal, and hardware-friendly framework called Non-uniform Linear Interpolation (NLI). NLI consists of two parts: NLI-Algorithm (software) and NLI-Engine (hardware). NLI-Algorithm replaces nonlinear evaluations with non-uniform interpolation in the FP16 domain. We formulate the cutpoint selection as a dynamic programming problem with an additive approximation objective that exhibits optimal substructure, enabling solution via the Bellman optimality principle. This yields globally optimal cutpoints and a calibration-free lookup table that is reusable across layers and models. NLI-Engine is a universal plug-and-play hardware block designed for nonlinear function computation based on the NLI algorithm. It improves computational efficiency by optimizing the underlying computation strategy. Software experiments demonstrate that NLI incurs negligible accuracy loss in the open-source LLMs Qwen and LLaMA, nor does it affect the accuracy of other DNN models. Hardware experiments demonstrate that the NLI Engine achieves more than 4× improvement in computational efficiency compared to the state-of-the-art designs. The main contributions presented are as follows:

- **Algorithm:** We introduce the **N**on-uniform **L**inear **I**nterpolation (NLI) framework, which casts cutpoint selection into a dynamic programming (DP) problem. Given a *fixed* nonlinear operator $f$ and a set of $N$ sorted candidate points in the FP16 domain, we minimize an *additive* interpolation error that sums per-segment costs over $M$ segments. Owing to the optimal substructure, the Bellman recursion solves for the *globally optimal* partition in $\mathcal{O}(MN^2)$ time. The resulting LUT is *calibration-free*—it depends only on $f$ and numeric settings rather than data distributions—and is therefore *reusable across layers and models*. We

provide a complete pipeline with implementation-ready interfaces and complexity-annotated pseudocode, facilitating immediate deployment with custom CUDA/Triton kernels.

- **Hardware Design:** We designed a general-purpose nonlinear computing circuit based on NLI engine. By leveraging a software-hardware co-design approach, we implemented a two-level address translation module to reduce the overhead of address conversion circuits. In addition, we employ pipelining to further boost throughput.

- **Comprehensive Experiment:** To demonstrate the practicality of NLI, we conduct experiments from both software and hardware perspectives. The software experiments show that our approximation strategy incurs negligible accuracy degradation in LLMs inference. Moreover, NLI exhibits strong generality, making it applicable not only to LLMs but also to other DNN models. We further synthesize the NLI engine using the SMIC 28nm technology and conduct a comprehensive analysis of its area, power, throughput and efficiency in comparison with SOTA designs.

## 2  BACKGROUND & RELATED WORK

### 2.1  NONLINEAR OPERATIONS OF LLMS

Mainstream LLMs such as LLaMA Touvron et al. (2023) and Qwen Bai et al. (2023) are typically composed of multi-head self-attention layers and feed-forward network (FFN) layers. Each self-attention layer includes one `Softmax` operation and one `RMSNorm` operation. Each FFN layer contains one `SiLU` operation and one `RMSNorm` operation. For MoE-based architectures such as DeepSeek-V3 Guo et al. (2024), each Transformer block typically contains multiple nonlinear operators—most commonly `Softmax` and `RMSNorm`—but can also incorporate `Sigmoid` for expert routing. Our approach remains broadly applicable here, as the `Sigmoid` operator can likewise be approximated through non-uniform linear interpolation, thereby offering a unified framework for various activations within MoE architectures. The definitions of these functions are as follows:

$$\text{RMSNorm}(x) = \frac{x}{\sqrt{\text{Mean}(x^2) + \epsilon}}, \quad \text{Softmax}(x_i) = \frac{e^{x_i - \max(x)}}{\sum_j e^{x_j - \max(x)}}, \quad \text{SiLU}(x) = \frac{x}{1 + e^{-x}}. \quad (1)$$

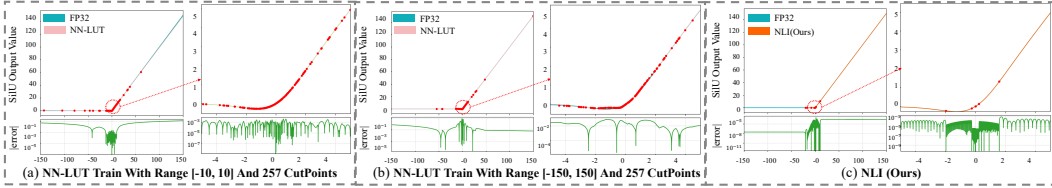

Figure 2: Approximation quality of the SiLU activation over the range $[-150, 150]$, which covers $\geq 99.9\%$ of activations under our measurement protocol (see Figure. 1 (a) and Appendix. A.4). Panel (a) and Panel (b) show the result of NN-LUT, and panel (c) shows our NLI framework. For each method we plot (top-left) the function curve, (bottom-left) the absolute error on a logarithmic scale, and (right) a zoom-in over [-5,5]. Red dots denote LUT cutpoints. NN-LUT suffers from pronounced error spikes, whereas NLI preserves near-machine-precision fidelity throughout the LLM-typical range, reducing worst-case error by several orders of magnitude.

As shown in the equations, these nonlinear operations contribute significantly to the overall runtime due to the high computational cost of floating-point arithmetic. Therefore, numerous approximation techniques have been proposed to alleviate the hardware costs. It is worth noting that earlier LLMs, such as OPT Zhang et al. (2022) and BLOOM Workshop et al. (2022), may have slight variations in their nonlinear operators compared to the equations above. However, the nonlinear functions they involve are still primarily based on exponential ($e^x$) and square root ($\sqrt{x}$) operations.

### 2.2  GENERAL-PURPOSE NONLINEAR COMPUTATION WORK

As a general-purpose NPU, it must be capable of handling a wide range of neural networks. Different types of neural networks utilize various nonlinear operators, such as `Softmax`, `Tanh`, and `ArcTan`.

While circuit-level optimization for a specific operator can achieve high efficiency and precision, it lacks flexibility and imposes significant limitations on broader applicability. Therefore, we attempt to approximate nonlinear functions using a general-purpose approach.

### 2.2.1 LINEAR LUT FITTING

Linear LUT Cantoni (1971); Karst (1958) approximation is a more generalized method for computing nonlinear functions. It approximates various nonlinear operators by storing N pairs of approximation parameters in a LUT. The computation formula is as follows:

$$\textbf{LinearLUT}(x) = \begin{cases} k_1 x + b_1 & \text{if } x < d_1, \\ k_i x + b_i & \text{if } d_{i-1} \le x < d_i, \quad \text{for } 1 < i \le N-1, \\ k_N x + b_N & \text{if } x \ge d_{N-1}. \end{cases} \tag{2}$$

Equation 2 shows that, for a fixed number of segments N, the approximation error is governed by the three linear parameters k, b, and d. NN-LUT Yu et al. (2022) extends LUT-Linear by modelling the search for $(k, b, d)$ with a Linear–ReLU–Linear network, whose piece-wise linearity allows the network weights to be analytically converted into the desired parameters. Although this strategy achieves high accuracy within the authors' test range, our re-implementation (the original code is not public) reveals a strong dependence on the span of the training data, leading to two major drawbacks: 1. **Severe extrapolation error**. When the training samples cover only a narrow interval—for SiLU, [-10,10]—the network generalises poorly outside that range, and its error explodes (Figure 2(a)). 2. **Poor convergence on wide ranges**. Expanding the interval to [-150,150], which covers 99.9% of the activations observed in mainstream LLMs, makes optimisation unstable; regions with high curvature (e.g., [-10,10]) cannot be fitted well, as shown in Figure 2(b). When the LUTs produced by NN-LUT are used to replace nonlinear operations in LLMs, model accuracy drops sharply (Figure 1(b)), underscoring that NN-LUT lacks generality for nonlinear functions with wide input ranges.

### 2.2.2 INTERPOLATION-BASED APPROXIMATIONS AND A UNIFIED ERROR BOUND

A standard strategy for approximating a nonlinear operator $f(\cdot)$ on resource-constrained accelerators is to replace it by an *interpolation algorithm* that reconstructs outputs from preselected support points using a local rule. Here, "interpolation algorithm" covers linear, piecewise polynomial, and spline-based variants.

**Piecewise polynomial and spline-base variants.** Polynomial interpolation and other higher-order Liu et al. (2015) interpolation methods have lower computational efficiency, so they are not suitable for NPUs. Previous works, such as NVDLA NVDLA (2017), have attempted to maintain accuracy by introducing an additional sub-table for regions where the function has steep variations, in addition to the base table. While this approach improves precision, it requires a two-level computation circuit, leading to lower efficiency. Moreover, for nonlinear functions with smooth slope variations, such as $Cos$, this method fails to achieve high accuracy.

**Linear interpolation** For a single segment $[a, b]$ with $f \in C^2([a, b])$, the exact (real-arithmetic) linear interpolant $P(x)$ admits the familiar worst-case bound. However, implementations evaluate a *finite-precision* approximation $\widetilde{P}(x)$ whose error includes both the analytical interpolation term and a rounding/quantization term. The two can be combined into a single bound:

$$\max_{x \in [a,b]} \left| f(x) - \widetilde{P}(x) \right| \le \underbrace{\frac{(b-a)^2}{8} \max_{x \in [a,b]} |f''(x)|}_{\text{interpolation term}} + \underbrace{\varepsilon_{\text{num}}}_{\text{finite-precision term}}, \tag{3}$$

where the first term is the classical linear–interpolation remainder and the finite-precision contribution can be bounded, to first order in the unit roundoffs, by

$$\varepsilon_{\text{num}} \le \Big( \gamma_1(u_c) + \gamma_2(u_a) \Big) \big( |y_i| + |y_{i+1}| \big), \qquad \gamma_k(u) = \frac{k\,u}{1 - k\,u}. \tag{4}$$

Here $y_i = f(a)$ and $y_{i+1} = f(b)$ are the endpoint values for the segment; $u_c$ models coefficient/storage precision (e.g., fp16 vs. fp32 for slopes/LUT values) and $u_a$ models arithmetic precision

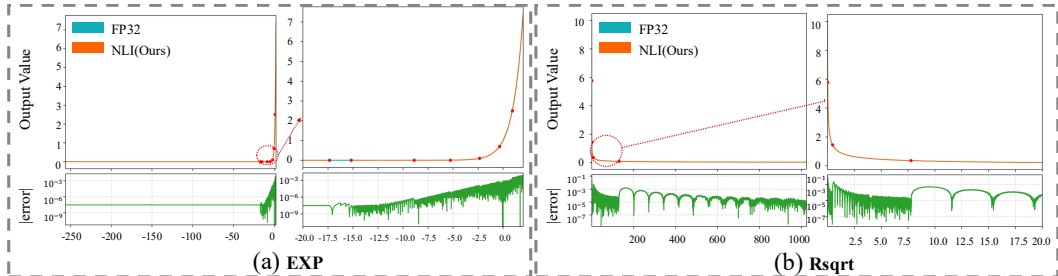

Figure 3: Comparison between the full-precision reference and our NLI approximation on two representative nonlinear functions—exp and rsqrt. We use 10 macro-intervals: the first and last are not subdivided, and each of the middle eight is uniformly partitioned into 32 bins; accounting for shared endpoints, this yields $2 + 32 \times 8 + 1 = 259$ cutpoints. The lower panels plot absolute error (log scale); the worst-case error stays below $1.2 \times 10^{-3}$ across the FP16 domain. Additional visualizations and cutpoint layouts are provided in Appendix A.6.

for the multiply–add evaluation. Equation 3 clarifies the accuracy levers of interpolation-based approximations: *(i)* the number of support points (more points shrink the segment length $(b-a)$), *(ii)* the distribution of points (allocating denser cuts where $|f''(x)|$ is large tightens the bound), and *(iii)* the storage and arithmetic precision (lower precision increases the total error via the finite-precision term $\varepsilon_{num}$). In practice, three widely used design heuristics fall short for modern LLMs. First, **uniform sampling** is intrinsically mismatched to functions with highly nonuniform curvature: since the worst-case linear–interpolation error on $[a, b]$ scales as $\frac{(b-a)^2}{8} \max |f''|$, uniform spacing leaves high-curvature regions under-resolved and wastes budget elsewhere; classical approximation theory recommends concentrating nodes in "difficult" regions (e.g., Chebyshev-like layouts) to mitigate Runge-type instability. Second, **curvature-driven point allocation** is not universally applicable: for many operators used in inference, the second derivative may be unavailable in closed form, whereas the numerical term $\varepsilon_{num}$ is typically left uncontrolled. Third, **training/calibration–based LUT fitting** ties accuracy to the span and distribution of calibration data; outside that span, extrapolation degrades and convergence becomes brittle. These limitations motivate a data-free, *globally optimal* placement of cutpoints under a fixed budget.

## 3 SOFTWARE METHODOLOGY

In this section, we present a *calibration-free*, and *globally optimal* (for a fixed knot budget) interval search algorithm for nonlinear functions, together with a more efficient hardware computation strategy.

### 3.1 PROPOSED NON-UNIFORM INTERPOLATION LUT VIA DYNAMIC PROGRAMMING

The discussion in Sec 2.2.2 (Equation. 3) shows that linear–interpolation error reflects both curvature and finite–precision terms, which explains why *uniform sampling*, *curvature–driven allocation*, and *training/calibration–based LUTs* each break down on LLMs. We therefore tackle the complementary design question: given a fixed budget of $M$ cutpoints (i.e., $M-1$ segments), where should the cutpoints be placed on the FP16 grid to minimize the global error actually seen by hardware? We cast this as a discrete dynamic program over the FP16 domain: the DP states and transitions (defined below) minimize the *mean relative error* on $\{x_0, \ldots, x_{N-1}\}$, include *endpoint clamping at both the first and last cutpoints* (the left clamp is captured in the $D[0, k]$ boundary term, and the right clamp as an explicit tail term), and are *calibration–free*.

**Setup.** We implement nonlinear operators via table lookup on the FP16 grid. Let $\mathcal{X} = \{x_0 < \cdots < x_{N-1}\}$ be all *finite* FP16 numbers that lie in the legal domain of $f$ (NaNs are dropped; $\pm\infty$ are clamped to the nearest finite endpoint). We choose $M$ cutpoints $\mathcal{B} = \{b_0, \ldots, b_{M-1}\} \subset \mathcal{X}$ with $b_0 = x_0$ and $b_{M-1} = x_{N-1}$, inducing $M-1$ macro-intervals. At inference time, inputs below $b_0$ (resp. above $b_{M-1}$) are clamped to $b_0$ (resp. $b_{M-1}$).

**Problem statement.** Given $f$, we seek $\mathcal{B}$ that minimizes the average *relative* interpolation error on the FP16 grid. Within each $[b_i, b_{i+1}]$ we approximate $f$ by the straight line through the *endpoints* $(b_i, f(b_i))$ and $(b_{i+1}, f(b_{i+1}))$.

**Rationale.** We optimize $M$ cutpoints over the sorted FP16 grid $\{x_0, \ldots, x_{N-1}\}$ for a target function $f$. Define two DP tables with explicit shapes and index ranges:

$$D \in \mathbb{R}^{M \times N}, \quad P \in \mathbb{Z}^{M \times N}, \qquad L \in \{0, \ldots, M-1\}, \; k \in \{0, \ldots, N-1\}.$$

Their meanings are:

- $D[L, k]$: the minimum error over the prefix $\{x_0, \ldots, x_k\}$ when $x_k$ is chosen as the $L$-th cutpoint.
- $P[L, k]$: the predecessor index (location of the $(L-1)$-th cutpoint) that attains $D[L, k]$.

**Error functional (mean relative error).** For any segment $[x_i, x_k]$ $(i < k)$, let $P_{i,k}(x)$ be the straight line through endpoints $(x_i, f(x_i))$ and $(x_k, f(x_k))$. We define

$$\text{Err}(i \to k) = \frac{1}{k - i + 1} \sum_{j=i}^{k} \frac{\big|f(x_j) - P_{i,k}(x_j)\big|}{\max\{|f(x_j)|, \tau\}},$$

where we set $\tau = 2^{-14}$, which equals the *smallest positive normal* value in IEEE 754 binary16 (FP16). This choice avoids over-amplifying relative errors for numerically near-zero activations while aligning the denominator floor with the FP16 normal/subnormal boundary.

**Boundary.** Inputs smaller than the first cutpoint are replaced by the first cutpoint's value. Hence, for the first cutpoint placed at $x_k$,

$$D[0, k] \;=\; \frac{1}{k+1} \sum_{j=0}^{k} \frac{\big|f(x_j) - f(x_k)\big|}{\max\{|f(x_j)|, \tau\}}, \qquad P[0, k] \;=\; k.$$

**Transition.** For $1 \le L \le M-1$ and $L \le k \le N-1$,

$$D[L, k] \;=\; \min_{i \in \{L-1, \ldots, k-1\}} \Big\{ D[L-1, i] \;+\; \text{Err}(i \to k) \;+\; \text{last\_error}(L, k) \Big\},$$

where the tail-clamping penalty is nonzero *only* at the last cutpoint:

$$\text{last\_error}(L, k) = \begin{cases} \dfrac{1}{\max\{1, \; N-1-k\}} \displaystyle\sum_{j=k+1}^{N-1} \frac{\big|f(x_j) - f(x_k)\big|}{\max\{|f(x_j)|, \tau\}}, & L = M - 1, \\ 0, & \text{otherwise.} \end{cases}$$

Set $P[L, k]$ to the argmin index $i$ that achieves $D[L, k]$.

**Optimal value.** The best $M$ cutpoints correspond to the minimum value in the last DP row:

$$\text{Cost}^* \;=\; \min_k D[M - 1, k].$$

**Backtracking.** Let $k^\star = \arg\min_k D[M - 1, k]$. Recover the indices of cutpoints by following predecessors:

$$\text{best\_points} = \big[k^\star, \; P[M - 1, k^\star], \; P[M - 2, P[M - 1, k^\star]], \; \ldots \big] \text{ (then reverse to ascending).}$$

**Computational cost.** The straightforward implementation is $\mathcal{O}(M \times N^2)$ (each state scans all predecessors). For typical settings ($M \le 11, N \le 63\,488$) the search completes in under ten minutes on a single NVIDIA RTX 4090 GPU with Triton.

The complete procedure is summarized in Algorithm 1. A naïve variant could set the DP budget to the *full* fine-grained table and search over $M{=}259$ cutpoints directly, but this inflates the DP time roughly $26\times$ (empirically $\approx 5$ hours on our setup) and would also mandate a large number of comparators in hardware, hurting throughput, area, and power.

Instead, we adopt a hardware-consistent layout with *ten* macro-intervals: the first and last are not subdivided, and each of the middle eight is uniformly partitioned into 32 bins. Under this layout, the DP only needs to optimize the *macro* endpoints, i.e., $M=11$ cutpoints, which reduces search time by about $26\times$ while producing LUTs that map directly to the two-level address translation. This design yields higher hardware efficiency (fewer comparators and smaller on-chip tables). Figure 2(c) and Figure 3 further visualise the resulting approximation quality: with only $2+8\times32+1 = 259$ cutpoints, our NLI overlaps almost perfectly with the FP32 curve, keeping the worst-case absolute error below $1.2 \times 10^{-3}$. Additional operators and cutpoint configurations are reported in Appendix A.6.

## 3.2 HARDWARE-FRIENDLY COMPUTATION STRATEGY

In this section, we propose a hardware-friendly computation strategy that utilizes two-level address translation. By leveraging simple computations, this approach significantly reduces the number of comparators and improves hardware efficiency.

Traditional address translation modules rely on multiple comparators. For example, with 259 cut points, the system generates 258 sub-intervals, requiring 259 parallel comparisons to determine the corresponding LUT address for an input data point. This leads to significant hardware overhead. To address this, we adopt a two-level address translation strategy. First, we divide the 259 cut points into a structure of $(2 + 8 \times 32 + 1)$, meaning 8 sub-intervals, each containing 32 cut points, along with two boundary values representing positive and negative infinity. Then, we use 10 comparators to determine which major interval the input data belongs to. Next, based on the interval index, we retrieve the multiply scale factor for that interval. After performing the multiplication, we apply a floor operation, and the resulting integer value serves as the LUT index. By utilizing a pipelined design, the latency introduced by the two-level approach is effectively hidden. Additionally, all multiply scale factors are precomputed offline and preloaded into dedicated registers. In this case, only 10 16-bit registers are required for storage. The detailed steps are shown in Algorithm 1.

As shown in Algorithm 2, the two-level address translation approach utilizes simple lookup and arithmetic operations. By applying the transformation $y = kx$ within each sub-interval, the address conversion is efficiently completed, eliminating the need for over 200 FP16 comparators. The computation formula for the linear interpolation can be summarized as follows:

$$y = Decimal \times (LUT[Index + 1] - LUT[Index]) + LUT[Index] \tag{5}$$

## 4 HARDWARE METHODOLOGY

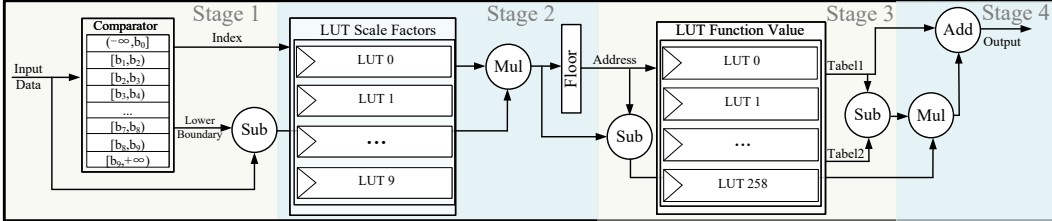

Figure 4: The hardware circuit design of a nonlinear computing unit, using two computing circuits sharing a set of Multiply scale factor and function value registers as an example.

In this section, we introduce the NLI Engine, a plug-and-play hardware module that leverages the optimized computation flow of the NLI algorithm to enable efficient nonlinear computation on NPUs.

Since current DNN accelerators fabricated with 28nm technology, e.g., Google TPU v1 Google (2016), operate at around 1GHz, we have adopted a four-stage pipeline design to align with the clock frequencies of most NPUs. The NLI engine implements the following functions: LUT and interval cutpoint preloading, two-level address translation, interpolation coefficient (Decimal) computation, and linear interpolation algorithm. The following are the detailed hardware design specifications of NLI engine:

**Stage 1 — Major-interval select & alignment.** An interval comparator (10 comparators) selects the macro-interval index $I \in \{0, \ldots, 9\}$ using preloaded left boundaries `left[I]`; inputs are clamped to $[\texttt{left}[0], \texttt{left}[9]]$. An FP16 subtractor computes the aligned offset $\Delta x = x - \texttt{left}[I]$.

**Stage 2 — Micro-address generation.** Each macro-interval has a preloaded scale $\mathtt{mul}[I]$ and a base pointer $\mathtt{base}[I]$ (start index in the global LUT). An FP16 multiplier forms $u = \Delta x \cdot \mathtt{mul}[I]$; the floor unit yields the integer micro-index $a = \lfloor u \rfloor$ and the fractional coefficient $t = u - a$ ($a \in [0, 31]$ for the middle eight intervals, $a = 0$ at the ends). The global LUT address is $g = \mathtt{base}[I] + a$.

**Stage 3 — Table read & slope prep.** A dual-port SRAM (259 entries) returns two adjacent values: $y_0 = \mathtt{LUT}[g]$ and $y_1 = \mathtt{LUT}[g + 1]$ in one cycle. An FP16 subtractor computes the local slope $\Delta y = y_1 - y_0$.

**Stage 4 — Linear interpolation.** An FP16 multiplier–adder evaluates $y = y_0 + t \cdot \Delta y$ (optionally fused FMA), then rounds to FP16.

This four-stage pipeline, together with the two-level address translation $(I, a)$, reduces comparators from 259 to 10 and shrinks address-translation overhead, improving throughput, area, and power.

## 5 EVALUATION

Table 1: NLI accuracy across datasets. Columns are datasets (higher is better except perplexity). Boldface rows highlight our method.

| Model | Method | Accuracy (↑) | | | | Perplexity (↓) |
|---|---|---|---|---|---|---|
| | | MMLU | GSM8k | HumanEval | Zero-shot Avg | Wikitext-2 |
| Llama3-8B | FP32 | 62.16 | 50.19 | 35.37 | 68.11 | 6.14 |
| | NN-LUT | 60.01 | 49.42 | 34.15 | 65.93 | 8.28 |
| | **NLI** | **62.14** | **50.49** | **35.37** | **68.24** | **6.14** |
| Llama3-70B | FP32 | 75.13 | 80.82 | 40.85 | 73.78 | 2.86 |
| | NN-LUT | 73.99 | 79.06 | 37.2 | 72.48 | 5.13 |
| | **NLI** | **75.11** | **81.27** | **40.63** | **73.85** | **2.86** |
| Qwen2.5-7B | FP32 | 70.56 | 44.28 | 40.24 | 67.48 | 7.46 |
| | NN-LUT | 25.51 | 0 | 0 | 30.13 | 28194 |
| | **NLI** | **70.67** | **43.97** | **39.63** | **67.63** | **7.46** |
| Qwen2.5-32B | FP32 | 81.74 | 70.13 | 56.71 | 70.76 | 5.32 |
| | NN-LUT | 25.51 | 0 | 0 | 30.70 | 70360 |
| | **NLI** | **81.68** | **70.07** | **55.88** | **70.67** | **5.32** |
| Qwen1.5-110B | FP32 | 79.26 | 84.44 | 50.84 | 72.42 | 4.81 |
| | NN-LUT | 75.99 | 76.15 | 43.03 | 69.08 | 6.83 |
| | **NLI** | **79.31** | **84.41** | **50.16** | **72.48** | **4.81** |
| Qwen3-8B | FP32 | 72.94 | 88.10 | 63.41 | 66.68 | 9.72 |
| | NN-LUT | 23.59 | 0 | 0 | 33.61 | 825.31 |
| | **NLI** | **72.98** | **88.17** | **62.59** | **66.76** | **9.73** |
| Qwen3-30B-A3B | FP32 | 77.86 | 85.44 | 31.21 | 67.59 | 8.70 |
| | NN-LUT | 28.46 | 0 | 0 | 65.60 | 10.76 |
| | **NLI** | **77.88** | **85.39** | **30.38** | **67.65** | **8.70** |

### 5.1 SOFTWARE EVALUATION

For evaluation, we first run NLI to search macro cutpoints for each nonlinear operator under a 10–macro-interval layout: the middle eight intervals are uniformly split into 32 bins, while the first/last are unsplit with endpoint clamping. We then replace nonlinear operators in Llama and Qwen (PyTorch Paszke (2019)) and report perplexity on Wikitext-2 Merity et al. (2016), a standard zero-shot suite (ARC-c/e Clark et al. (2018), BoolQ Clark et al. (2019), PIQA Bisk et al. (2020), HellaSwag Zellers et al. (2019), OBQA Mihaylov et al. (2018), LAMBADA Paperno et al. (2016), SIQA Sap et al. (2019), WinoGrande Sakaguchi et al. (2021)), and three widely used benchmarks—MMLU Hendrycks et al. (2020) (broad factual/problem-solving across 57 disciplines), HumanEval Chen et al. (2021) (functional code-generation accuracy), and GSM8k Cobbe et al. (2021) (multi-step grade-school math). We cover multiple model scales and, to test generality beyond LLMs, repeat the substitution on ViT and CNNs.

From Table 1, we observe that replacing nonlinear operators with NLI in Llama and Qwen yields *no accuracy drop* on the zero-shot suite (detailed results in Appendix A.5.2) and does not increase PPL. Moreover, performance on MMLU, HumanEval, and GSM8k remains nearly on par with FP32. Our nonlinear computation strategy has minimal impact on the accuracy of open-source Llama/Qwen models—even without any data calibration. Beyond LLMs, substituting NLI for nonlinear operators in ViT and representative CNNs yields no statistically significant accuracy degradation; full per-model results are reported in Appendix A.5.1 (Table 7).

### 5.1.1 ABLATION

Table 2: Ablation I on Qwen2.5-7B: two-level NLI (259) vs macro-only (11).

| Method | Cutpoints | MMLU | GSM8k |
|---|---|---|---|
| FP32 | – | 70.56 | 44.28 |
| NLI (2+8×32+1) | 259 | **70.67** | **43.97** |
| Macro-only (DP, $M=11$) | 11 | 21.14 | 0 |

Table 3: Ablation II on Qwen2.5-7B: accuracy (↑) and cutpoint-search time (↓).

| Method | Cutpoints | MMLU | GSM8k | Search time(s) |
|---|---|---|---|---|
| FP32 | – | 70.56 | 44.28 | – |
| NLI (2+8×32+1) | 259 | **70.67** | 43.97 | 610 |
| Non-uniform 259 (DP) | 259 | 70.65 | **44.08** | 17000 |

As shown in Table 2, we evaluate on Qwen2.5-7B with zero-shot MMLU and GSM8K. Compared to the proposed two-level layout **NLI** (259 total cutpoints), the *macro-only* variant optimizes only $M=11$ endpoints and applies a single piecewise-linear fit over the FP16 grid (with no per-macro uniform sub-bins). This ablation isolates the benefit of uniform micro-partitioning within each macro-interval— demonstrating that using only 11 cutpoints is insufficient on both benchmarks.

We compare **NLI** with a direct DP over **259 non-uniform** cutpoints (no macro/micro constraint) on **Qwen2.5-7B**. As shown in Table 3, accuracy on MMLU/GSM8k is essentially unchanged, while the search time explodes ($\sim 28\times$ slower) and the resulting layout is hardware-unfriendly, incurring higher area/latency costs.

We further compare **NLI** with two common heuristics using the same total budget of 259 points: (i) **Uniform 259** (uniformly spaced over the FP16 grid), and (ii) **Curvature 259** (density proportional to a curvature proxy). Both use the same linear interpolation and inference pipeline as NLI. Table 4 summarizes results.

Table 4: Ablation III on Qwen2.5-7B: accuracy comparison on MMLU and GSM8k.

| Method | Cutpoints | MMLU | GSM8k |
|---|---|---|---|
| FP32 | – | 70.56 | 44.28 |
| NLI (2+8×32+1) | 259 | **70.65** | **43.97** |
| Uniform 259 | 259 | 45.91 | 18.13 |
| Curvature 259 | 259 | 65.74 | 32.58 |

### 5.2 HARDWARE EVALUATION

The experiments in the previous sections have demonstrated that our NLI approximation method achieves high accuracy and generality. In this section, to demonstrate the efficiency of the NLI engine, we compare NLI engine with two state-of-the-art nonlinear computational units.

We implemented the hardware circuit design using Chisel Bachrach et al. (2012). The circuit was synthesized with Design Compiler Muchnick (1997) under the SMIC 28nm process library to obtain area , power, and timing information.

Table 5: Hardware area breakdown.

| | LUT | Comparator | Multiplier | Adder | Others | Total ($\mu m^2$) |
|---|---|---|---|---|---|---|
| NN-LUT | 12268 | 10496 | 205 | 134 | 135 | 23238 |
| RI-LUT | 12268 | 10496 | 410 | 268 | 205 | 23647 |
| **NLI** | **6445** | **410** | **205** | **536** | **191** | **7787** |

Table 6: Hardware comparison.

| | Clock Freq. | Area ($\mu m^2$) | Power ($mW$) | Throughput | Efficiency |
|---|---|---|---|---|---|
| NN-LUT | 1GHz | 23238 | 46 | 1G | 0.94 |
| RI-LUT | 1GHz | 23647 | 48 | 1G | 0.88 |
| **NLI** | **1GHz** | **7787** | **34** | **1G** | **3.78** |

Table 5 shows the areas of LUTs, multipliers, adders, and Others (registers, shifters, etc.) in the three general nonlinear computational units: NLI, NN-LUT, and RI-LUT. It can be seen that NLI (with 259 cut points) saves 68% and 69% in area compared with the other two SOTAs (NN-LUT and RI-LUT, each with 256 cut points).

NN-LUT Yu et al. (2022) and RI-LUT Kim et al. (2023) require storing $512 \times 16$ bits of data (256 K values and 256 B values). NLI only needs to store data for 259 16-bit cut points and 10 16-bit scale factors , so it has a smaller LUT area. Also, our hardware-friendly algorithm cuts down the number of comparators. NN-LUT and RI-LUT need 256 comparators to divide into 256 interval addresses, while NLI only needs 10 comparators for ten intervals.

Table 6 provides a detailed comparison of the NLI Engine, NN-LUT, and RI-LUT hardware modules under a 1 GHz clock frequency in terms of area, power, Throughput ( the number of nonlinear operators computed per cycle $\times$ clock frequency), and Efficiency ($Throughput/(area \times power)$). The NLI Engine exhibits lower power consumption, primarily due to reduced static power resulting from fewer LUTs and comparator modules. Since all three hardware units employ pipelining, each cycle produces one completed result once the pipeline is filled. Therefore, the throughput of all three modules is 1G. Benefiting from its lower area and power consumption, the NLI Engine achieves 4.02× higher efficiency than NN-LUT and 4.29× higher efficiency than RI-LUT.

## 6 CONCLUSION

In this paper, we propose **NLI**, a non-uniform linear interpolation method for efficiently approximating nonlinear functions in large language models. By formulating cutpoint selection as a dynamic programming problem, NLI achieves near-optimal accuracy with minimal hardware overhead. Experiments demonstrate that NLI maintains model accuracy without calibration, generalizes well across diverse models, and significantly reduces hardware resource usage compared to state-of-the-art methods. We believe NLI provides a practical solution for deploying large models efficiently on resource-limited hardware.

## ETHICS STATEMENT

This work does not involve human subjects, personal data, or sensitive content. It focuses solely on algorithmic and hardware-level optimization of nonlinear function approximation. Therefore, we believe it does not raise ethical concerns.

## REPRODUCIBILITY STATEMENT

We provide complete details of our algorithms, hyperparameters, and evaluation protocols in the main paper and appendix. All models are evaluated on publicly available benchmarks (Wikitext-2, ARC, BoolQ, PIQA, HellaSwag, OBQA, LAMBADA, SIQA, WinoGrande, MMLU, HumanEval, GSM8k, and standard vision datasets). The code for constructing lookup tables, performing DP cutpoint search, and reproducing our experiments will be released upon publication. These resources will ensure full reproducibility of the reported results.

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

## A APPENDIX

### A.1 DP-OPTIMAL MACRO CUTPOINT SEARCH

---

**Algorithm 1:** DP-Optimal Macro Cutpoint Search (NLI; mean relative error with endpoint clamping)

---

**Input:** Sorted FP16 grid $\mathcal{X} = \{x_0, \ldots, x_{N-1}\}$; target function $f(\cdot)$; number of *cutpoints* $M \geq 2$; small constant $\tau > 0$.

**Output:** Optimal cutpoints $\mathcal{B} = \{b_0, \ldots, b_{M-1}\}$ with $b_L \in \mathcal{X}$ and their values $\{f(b_L)\}$.

*1. Precompute values*
> **for** $k \leftarrow 0$ **to** $N - 1$ **do**
> > $y_k \leftarrow f(x_k)$

*2. Define error functionals*
> // Mean *relative* error of the endpoint-anchored line on $[i,k]$
>
> 1   $\texttt{Err}(i,k) \triangleq \dfrac{1}{k-i+1} \sum_{j=i}^{k} \dfrac{\left| y_j - P_{i,k}(x_j) \right|}{\max\{|y_j|, \tau\}}, \quad P_{i,k}(x) = y_i + \dfrac{y_k - y_i}{x_k - x_i}(x - x_i) \; ;$
>
> // Right-end clamping penalty for the last cutpoint (zero if $k = N-1$)
>
> 2   $\texttt{TailClamp}(k) \triangleq \begin{cases} \frac{1}{N-1-k} \sum_{j=k+1}^{N-1} \frac{|y_j - y_k|}{\max\{|y_j|, \tau\}}, & k < N-1, \\ 0, & k = N-1, \end{cases}$

*3. Initialize DP tables*
> // $d \in \mathbb{R}^{M \times N}$ stores minimal prefix cost; $p \in \mathbb{Z}^{M \times N}$ stores predecessors
> Initialize $d[L,k] \leftarrow +\infty$, $p[L,k] \leftarrow -1$ for all $L, k$;
> // Left-end clamping: first cutpoint at $x_k$ uses constant $y_k$ on $[0,k]$
> **for** $k \leftarrow 0$ **to** $N - 1$ **do**
> > $d[0,k] \leftarrow \frac{1}{k+1} \sum_{j=0}^{k} \frac{|y_j - y_k|}{\max\{|y_j|, \tau\}};$
> > $p[0,k] \leftarrow k;$

*4. Fill DP (macro endpoints)*
> **for** $L \leftarrow 1$ **to** $M - 1$ **do**
> > **for** $k \leftarrow L$ **to** $N - 1$ **do**
> > > best $\leftarrow +\infty$;   arg $\leftarrow -1$;
> > > **for** $i \leftarrow L - 1$ **to** $k - 1$ **do**
> > > > val $\leftarrow d[L-1, i] + \texttt{Err}(i,k) + \mathbf{1}_{\{L=M-1\}} \cdot \texttt{TailClamp}(k);$
> > > > **if** *val < best* **then**
> > > > > best $\leftarrow$ val;;
> > > > > arg $\leftarrow i$
> > >
> > > $d[L,k] \leftarrow$ best;     $p[L,k] \leftarrow$ arg;

*5. Backtrack optimal cutpoints*
> $k^\star \leftarrow \arg\min_{k \in \{M-1, \ldots, N-1\}} d[M-1, k];$
> // Recover indices of $M$ cutpoints, from last to first
> $\text{idx}[M-1] \leftarrow k^\star;$
> **for** $L \leftarrow M - 1$ **down to** $1$ **do**
> > $\text{idx}[L-1] \leftarrow p[L, \text{idx}[L]];$
>
> **for** $L \leftarrow 0$ **to** $M - 1$ **do**
> > $b_L \leftarrow x_{\text{idx}[L]};$    $f(b_L) \leftarrow y_{\text{idx}[L]};$
>
> **return** $\mathcal{B} = \{b_0, \ldots, b_{M-1}\}$ *and* $\{f(b_L)\}_{L=0}^{M-1};$

---

## A.2 NLI COMPUTATION FLOW

---

**Algorithm 2:** NLI Computation Flow

---

**Input:** FP16 input $x$; interval endpoints $Point[0{:}10]$ (11 points); per-interval scales $Mul[0{:}9]$
(10 values); LUT values $LUT[0{:}258]$ (259 points); uniform bins per macro-interval
$D_n{=}32$ for intervals $1\ldots 8$.
**Output:** Nonlinear output $y$.
**Constants:** $M{=}10$ intervals; indices are 0-based.;

**Preload registers:** $Point\_Reg \leftarrow Point$, $Mul\_Reg \leftarrow Mul$, $LUT\_Reg \leftarrow LUT$.;
 `// one-time load`

**Clamp input to domain:** $x \leftarrow \text{clip}\big(x,\ Point\_Reg[0],\ Point\_Reg[10]\big).$;

**Locate interval** $Index \in \{0, \ldots, 9\}$: **for** $i \leftarrow 0$ **to** $9$ **do**
 **if** $Point\_Reg[i] \leq x < Point\_Reg[i{+}1]$ **then**
  $Index \leftarrow i$; **break**
`// Equivalently, this can be done via bucketize.`

**Local coordinate in interval** $Index$**:**
$Temp \leftarrow x - Point\_Reg[Index]$;   `// offset within the interval`
$Mul\_Temp \leftarrow Temp \times Mul\_Reg[Index]$;    `// scaled position`
$Address \leftarrow \lfloor Mul\_Temp \rfloor$; `// integer bin (0 for intervals 0/9; 0..31`
 `for 1..8)`
$Decimal \leftarrow Mul\_Temp - Address$;  `// fractional part within the bin`

**Global LUT index** $indices$:

$$indices \leftarrow \begin{cases} 0 + Address, & \text{if } Index = 0, \\ 1 + (Index{-}1)\cdot D_n + Address, & \text{if } Index \geq 1. \end{cases}$$

**Linear interpolation from LUT:**
$Left \leftarrow LUT\_Reg[indices], \quad Right \leftarrow LUT\_Reg[indices{+}1]$
$y \leftarrow Left + Decimal \times (Right - Left)$

**Boundary saturation:**
**if** $x \leq Point\_Reg[0]$ **then** $y \leftarrow LUT\_Reg[0]$;
**if** $x \geq Point\_Reg[10]$ **then** $y \leftarrow LUT\_Reg[258]$;
**return** $y$

---

## A.3 LLM USAGE DECLARATION

We disclose that large language model (LLM) tools were used *only* for language editing—copy-editing, stylistic smoothing, and minor rephrasing of prose—and limited formatting assistance. No LLM was involved in conceiving the research, proposing methods, generating technical content, writing code, running experiments, analyzing data, or drawing conclusions.

All substantive elements of this work were completed by the authors, including:

- problem formulation, research design, and overall narrative structure;

- algorithmic development, theoretical reasoning/derivations, implementation, and debugging;

- experimental design, data collection and preprocessing, execution, evaluation, and interpretation of results.

Any text suggestions produced by LLM tools were reviewed and edited by the authors. The authors accept full responsibility for the scientific validity, accuracy, and originality of the manuscript and affirm adherence to standards of academic integrity.

A.4   ACTIVATION COVERAGE MEASUREMENT PROTOCOL

**Models and operators.**    We measure the pre-activation inputs to nonlinear operators (e.g., SiLU, RMSNorm, Softmax) across the LLMs listed in Figure. 1 (a).

**Data and setup.**    To align with our main evaluation, we use the same public corpora employed in Sec. 5 (e.g., Wikitext-2 for perplexity and the zero-shot suite for accuracy). All models are run in FP16 inference with the same tokenization and maximum sequence length as in our evaluation.

**Aggregation across layers and tokens.**    For each model, we collect activations from all layers over all tokens. We then compute the 0.05th and 99.95th percentiles of the resulting distribution. To form a symmetric interval, we set $r_{model} = \max\{|q_{0.0005}|, |q_{0.9995}|\}$ and define the model-specific range as $[-r_{model}, r_{model}]$.

**Cross-model coverage.**    We aggregate models by taking the union of their symmetric ranges, i.e., $r_{max} = \max_{model} r_{model}$, and define the LLM-typical domain as $[-r_{max}, r_{max}]$. In our measurements, $r_{max} \leq 150$, hence $[-150, 150]$ *covers* $\geq$*99.9% of observed activations.*

**Outliers.**    Values outside this domain account for $\leq 0.1\%$ of activations and are clamped at runtime by our engine; they have negligible impact on end-to-end accuracy.

A.5   FULL EXPERIMENT RESULTS.

A.5.1   SUPPLEMENTARY EVALUATION ON NON-LLM TASKS

Table 7: Accuracy performance of NLI on vision models

| Model | Method | mAP (%) (obj det) | Top-1 Acc (%) (cls) |
|---|---|---|---|
| DETR | FP32 | 39.4 | — |
| | **NLI** | **39.4** | — |
| ViT-Small | FP32 | — | 74.7 |
| | **NLI** | — | **74.7** |
| RT-DETR-L | FP32 | 52.5 | — |
| | **NLI** | **52.5** | — |
| YOLOv8-M | FP32 | 50.1 | — |
| | **NLI** | **50.1** | — |

To further verify the generality of NLI beyond LLMs, we replaced the nonlinear operators in representative vision models, including YOLOv8 Varghese & Sambath (2024), DETR Carion et al. (2020), ViT Dosovitskiy et al. (2020), and RT-DETR Zhao et al. (2024). As shown in Table 7, employing NLI for nonlinear operator computation does not incur any accuracy degradation, demonstrating that the proposed framework is broadly applicable across diverse architectures and modalities.

A.5.2   FULL ZERO-SHOT EVALUATION ON LLMS

Table 8 reports the complete zero-shot evaluation results of NLI, NN-LUT, and FP32 baselines on all considered LLMs. Compared with NN-LUT, NLI consistently preserves baseline-level accuracy across tasks and scales, while avoiding the severe degradation observed with NN-LUT.

A.6   ADDITIONAL NLI VISUALISATIONS AND FP16 CUTPOINTS TABLES

To demonstrate the breadth and stability of **NLI** across common nonlinear operators, Figure 5 visualises fits for eight functions widely used in modern models: `exp`, `gelu`, `rsqrt`, `reciprocal`, `hardswish`, `mish`, `sigmoid`, and `tanh`. Each panel contains two subplots: the left shows the full FP16-domain behaviour (top: FP32 reference in teal and NLI in orange; bottom: absolute error on a log scale), and the right provides a zoom-in around the high-curvature region that typically

Table 8: NLI accuracy on multiple LLMs. Boldface rows highlight our method.

| Model | Method | Zero-Shot (↑) | | | | | | | | | | Perplexity (↓) |
| | | ARC-c | ARC-e | BoolQ | PIQA | HellaS | OBQA | Lam. | SIQA | WinoG. | Avg. | Wikitext-2 |
|---|---|---|---|---|---|---|---|---|---|---|---|---|
| Llama3-8B | FP32 | 53.16 | 77.78 | 81.25 | 80.74 | 79.10 | 44.80 | 75.66 | 47.08 | 73.40 | 68.11 | 6.14 |
| | NN-LUT | 50.11 | 75.21 | 79.18 | 78.93 | 77.66 | 42.60 | 75.07 | 45.47 | 71.24 | 65.93 | 8.28 |
| | **NLI** | **53.67** | **77.78** | **81.41** | **80.74** | **79.23** | **44.80** | **75.70** | **47.08** | **73.72** | **68.24** | **6.14** |
| Llama3-70B | FP32 | 64.25 | 86.03 | 85.29 | 84.44 | 85.00 | 48.60 | 79.35 | 50.67 | 80.43 | 73.78 | 2.86 |
| | NN-LUT | 63.16 | 84.99 | 84.06 | 82.69 | 83.51 | 46.16 | 78.97 | 50.01 | 78.76 | 72.48 | 5.13 |
| | **NLI** | **64.33** | **86.07** | **85.08** | **84.39** | **84.92** | **49.00** | **79.41** | **50.67** | **80.74** | **73.85** | **2.86** |
| Qwen2.5-7B | FP32 | 51.28 | 76.47 | 85.96 | 78.67 | 79.55 | 48.00 | 67.67 | 50.36 | 69.38 | 67.48 | 7.46 |
| | NN-LUT | 19.62 | 25.38 | 37.83 | 52.12 | 25.61 | 28.20 | 0.00 | 33.01 | 49.41 | 30.13 | 28194 |
| | **NLI** | **51.37** | **76.39** | **85.96** | **79.82** | **79.46** | **48.20** | **67.67** | **50.61** | **69.22** | **67.63** | **7.46** |
| Qwen2.5-32B | FP32 | 58.36 | 77.19 | 89.69 | 81.39 | 85.25 | 46.00 | 75.26 | 50.05 | 73.64 | 70.76 | 5.32 |
| | NN-LUT | 20.99 | 23.95 | 38.63 | 53.92 | 26.64 | 28.80 | 0.00 | 33.21 | 50.12 | 30.70 | 70360 |
| | **NLI** | **58.28** | **77.44** | **89.72** | **81.23** | **85.18** | **45.40** | **75.20** | **50.15** | **73.40** | **70.67** | **5.32** |
| Qwen1.5-110B | FP32 | 55.46 | 76.94 | 88.90 | 83.84 | 86.13 | 46.60 | 78.52 | 54.35 | 81.06 | 72.42 | 4.81 |
| | NN-LUT | 52.01 | 74.08 | 83.87 | 80.32 | 83.77 | 42.40 | 75.99 | 51.32 | 77.96 | 69.08 | 6.83 |
| | **NLI** | **55.80** | **77.10** | **88.96** | **83.90** | **86.11** | **46.80** | **78.36** | **54.25** | **81.06** | **72.48** | **4.81** |
| Qwen3-8B | FP32 | 55.29 | 80.93 | 86.70 | 77.69 | 74.92 | 41.40 | 64.04 | 51.84 | 67.32 | 66.68 | 9.72 |
| | NN-LUT | 23.01 | 28.99 | 41.65 | 57.16 | 30.13 | 29.71 | 4.90 | 36.14 | 50.79 | 33.61 | 825.31 |
| | **NLI** | **55.29** | **80.93** | **86.70** | **77.90** | **75.11** | **41.53** | **64.21** | **51.79** | **67.36** | **66.76** | **9.73** |
| Qwen3-30B-A3B | FP32 | 52.30 | 79.38 | 88.59 | 79.43 | 76.68 | 45.00 | 64.84 | 51.23 | 69.85 | 67.59 | 8.70 |
| | NN-LUT | 50.71 | 77.86 | 87.03 | 77.69 | 75.23 | 44.13 | 60.00 | 49.76 | 68.01 | 65.60 | 10.76 |
| | **NLI** | **52.24** | **79.58** | **88.57** | **79.51** | **77.68** | **44.98** | **64.91** | **51.46** | **69.94** | **67.65** | **8.70** |

dominates the global error budget. Red dots mark the LUT cutpoints produced by our $2+8 \times 32+1$ budget (top-level endpoints plus uniformly spaced interior cuts). Across all eight operators, NLI tracks the FP32 curve almost perfectly; the worst-case absolute error remains within $1.5 \times 10^{-3}$ over the entire domain and is orders of magnitude smaller in most subranges.

Table 9 lists the *top-level* FP16 cutpoint endpoints (in decimal) that define 10 macro-intervals for each operator. The full lookup table with $2 + 8 \times 32 + 1 = 259$ entries is obtained by placing 32 uniformly spaced cut points inside each macro-interval and adding two boundary values. During inference, FP16 inputs that fall below the smallest endpoint or above the largest endpoint are *clamped* to the respective boundary before lookup, which guarantees numerical stability for extreme values (e.g., very small arguments in `rsqrt` and `reciprocal`). For completeness, we also release the exact LUTs used in our experiments.

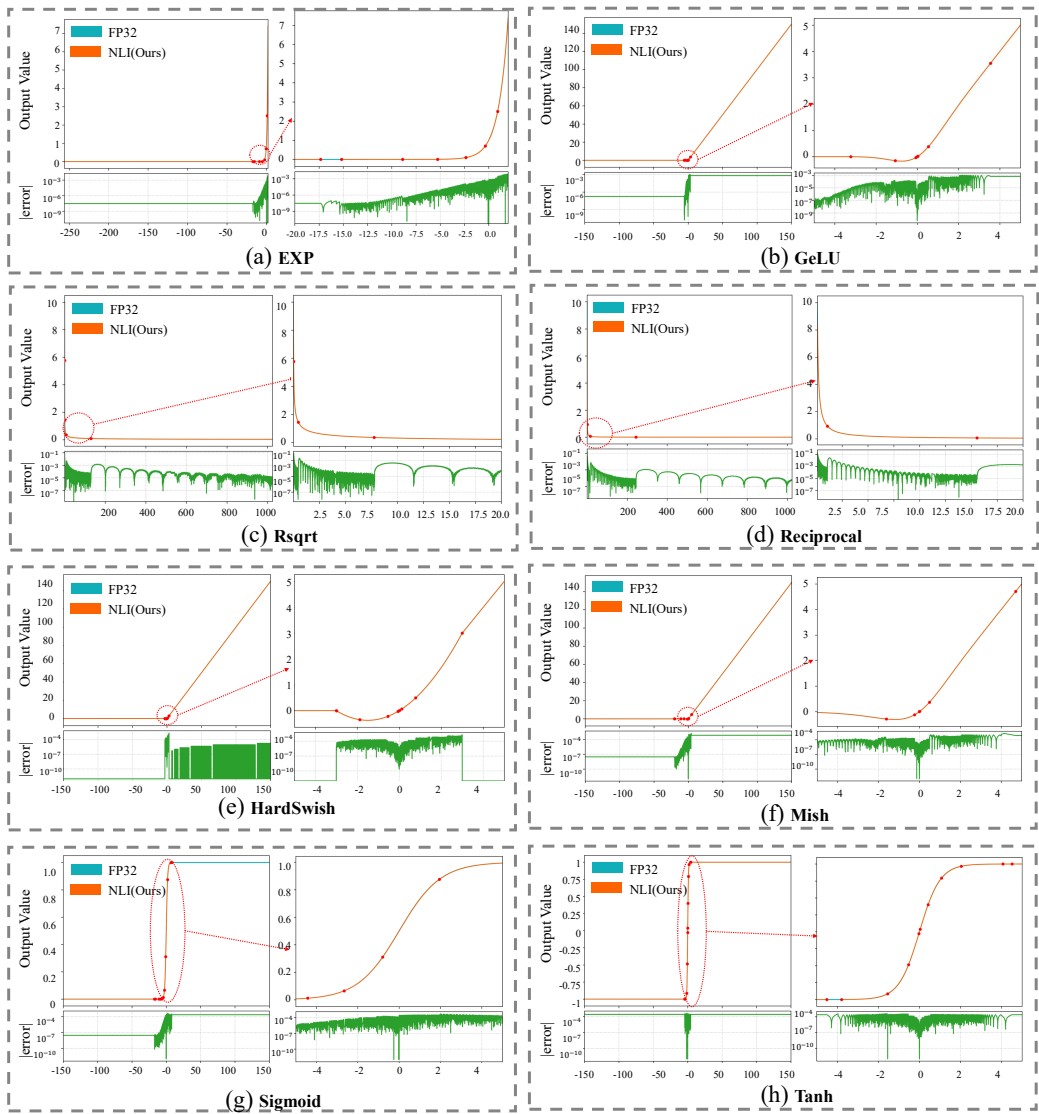

Figure 5: NLI approximation quality for eight representative nonlinear operators: (a) exp, (b) gelu, (c) rsqrt, (d) reciprocal, (e) hardswish, (f) mish, (g) sigmoid, and (h) tanh. For each operator, the left subplot shows the full-domain fit (top: FP32 reference vs. NLI; bottom: absolute error in log scale), while the right subplot zooms into the high-curvature region. Red dots denote LUT cutpoints generated under an $2+8 \times 32+1$ budget. NLI closely overlaps the FP32 reference, keeping the worst-case absolute error within $1.5 \times 10^{-3}$ across the FP16 domain.

| Function | Range | #Segments | Top-level cutpoints (FP16, decimal) |
|---|---|---|---|
| gelu | $[-5.5390625, 65504.0]$ | $2+8\times32+1$ | -5.5390625, -5.15625, -3.18359375, -0.98046875, -0.1229248046875, -0.00374603271484375, 0.0035247802734375, 0.11322021484375, 0.78076171875, 4.10546875, 65504.0 |
| silu | $[-20.359375, 65504.0]$ | $2+8\times32+1$ | -20.359375, -17.109375, -8.3671875, -1.9755859375, -0.255615234375, -0.007244110107421875, 0.0072174072265625, 0.228515625, 1.58203125, 10.46875, 65504.0 |
| exp | $[-17.34375, 11.0859375]$ | $2+8\times32+1$ | -17.34375, -15.171875, -8.890625, -5.2734375, -2.35546875, -0.3583984375, 0.91650390625, 3.451171875, 6.84765625, 10.9453125, 11.0859375 |
| reciprocal | $[1.5318394\times10^{-5}, 65504.0]$ | $2+8\times32+1$ | 1.5318394e-05, 2.2590160e-05, 4.6992302e-04, 7.0533752e-03, 8.8378906e-02, 1.07421875, 15.546875, 244.5, 3694.0, 46560.0, 65504.0 |
| rsqrt | $[5.9604645\times10^{-8}, 65504.0]$ | $2+8\times32+1$ | 5.9604645e-08, 7.7486038e-07, 1.1140108e-04, 1.8644333e-03, 3.0029297e-02, 0.48193359375, 7.7734375, 129.75, 2406.0, 47456.0, 65504.0 |
| hardswish | $[-3.0, 65504.0]$ | $2+8\times32+1$ | -3.0, -2.984375, -1.87890625, -0.5390625, -0.059326171875, -0.000743865966796875, 0.0034942626953125, 0.11968994140625, 0.78369140625, 3.001953125, 65504.0 |
| tanh | $[-4.5078125, 4.5078125]$ | $2+8\times32+1$ | -4.5078125, -3.79296875, -1.55078125, -0.5302734375, -0.028564453125, 0.0364990234375, 0.423828125, 1.076171875, 2.0390625, 4.0625, 4.5078125 |
| mish | $[-20.34375, 65504.0]$ | $2+8\times32+1$ | -20.34375, -19.90625, -10.921875, -6.2265625, -1.615234375, -0.237060546875, -0.00699615478515625, 0.01538848876953125, 0.491455078125, 4.70703125, 65504.0 |
| sigmoid | $[-17.34375, 8.3203125]$ | $2+8\times32+1$ | -17.34375, -15.765625, -10.65625, -8.15625, -6.3046875, -4.421875, -2.6640625, -0.7998046875, 1.9462890625, 6.90234375, 8.3203125 |

Table 9: Top-level cutpoint endpoints used by NLI. Each row lists 11 FP16 cutpoints (decimal) that define 10 macro-intervals; placing 32 uniformly spaced cut points inside each macro-interval plus two boundary values yields a total of $2+8\times32+1 = 259$ entries. *Inference uses clamping: FP16-domain inputs below the smallest cutpoint or above the largest cutpoint are clipped to the respective endpoint.*

