# OpenReview forum: "NLI : Non-uniform Linear Interpolation Approximation of Nonlinear Operations for Efficient LLMs Inference"
_ICLR.cc/2026/Conference — ICLR 2026 Poster_

### Official Review · Reviewer_KggQ · 2025-10-27

**Soundness:** 1
**Presentation:** 4
**Contribution:** 3
**Rating:** 2
**Confidence:** 4

**Summary:**

This paper proposes NLI, a software–hardware co-design for efficient approximation and computation of nonlinear operations in LLM inference. Specifically, the NLI framework performs non-uniform interpolation of nonlinear functions, formulating the cutpoint selection as a DP problem. In addition, NLI introduces a dedicated hardware engine for it that can be seamlessly integrated into existing LLM inference hardware.

**Strengths:**

* The analysis of the limitations of existing approaches (e.g., NN-LUT) for LLM inference is insightful and interesting.
* The proposed solution, encompassing both algorithmic and hardware aspects, appears technically sound, and the evaluations are thorough and convincing.

**Weaknesses:**

The motivation of this work is not sufficiently persuasive, which makes the overall contribution difficult to fully appreciate. While the authors emphasize the computational complexity of nonlinear operations in LLMs, I believe their actual overhead constitutes only a small fraction of the end-to-end inference time, since linear operations remain dominant even after various optimizations or compression techniques. Moreover, it is questionable whether the hardware cost of special function units in existing GPUs or NPUs is significant enough to justify the need for this entire line of research.

**Questions:**

Do we really need to optimize nonlinear operations in LLMs? This is my primary and most critical question regarding this paper. I would like to see both quantitative and qualitative discussions on the actual performance and hardware overheads associated with nonlinear operations, to better justify the necessity of this research direction.

---

> ### Author Response · Authors · 2025-11-22
> **Response to Reviewer KggQ**
>
> Dear Reviewer KggQ,
>
> We accurately accept your criticism regarding the motivation of our work and sincerely thank you for the "wake-up call." We fully understand why the necessity of optimizing nonlinear operations might seem questionable at first glance, given that their FLOPs count is indeed negligible compared to linear projections. **Your feedback identified a critical gap in our original manuscript:** we failed to provide sufficient quantitative evidence linking nonlinear operations to actual *hardware bottlenecks* (throughput and area) rather than just arithmetic complexity.
>
> To address your primary concern—which challenges the fundamental premise of our research—we have conducted extensive additional profiling on both H100 GPUs and edge NPUs. We believe the detailed breakdown below will demonstrate that despite the low FLOPs, nonlinear operations constitute a significant and growing latency bottleneck in modern LLM inference.
>
>
> ---
> ```
> W1: The motivation of this work is not sufficiently persuasive, which makes the overall contribution difficult to fully appreciate. While the authors emphasize the computational complexity of nonlinear operations in LLMs, I believe their actual overhead constitutes only a small fraction of the end-to-end inference time, since linear operations remain dominant even after various optimizations or compression techniques. Moreover, it is questionable whether the hardware cost of special function units in existing GPUs or NPUs is significant enough to justify the need for this entire line of research.
> ```
> **We agree that the arithmetic complexity (FLOPs) is low, but the real bottleneck lies in Hardware Throughput and Area Efficiency.**
>
> 1.  **GPU Throughput Gap (The SFU Wall):** Modern GPUs heavily favor linear computation. On NVIDIA H100, the FP16 Tensor Core throughput (989 TFLOPS) is **~250$\times$** higher than the Special Function Unit (SFU) throughput (3.9 TFLOPS).
>     * **Latency Reality:** Due to this disparity, for a standard head dimension ($D=128$), the execution time of `exp` is approximately **half** that of the Attention Matrix Multiplication ($T_{\exp} \approx 0.5 \times T_{\text{matmul}}$), despite negligible FLOPs.
>     * **Scaling Trend:** As linear layers are further accelerated by INT8/INT4 kernels, nonlinear operations (which rely on slow SFUs) become the dominant tail-latency source.
>
> 2.  **NPU Area Constraints (The Cost Wall):** Nonlinear units are disproportionately expensive in silicon area.
>     * **Area Bottleneck:** Based on SMIC 28nm synthesis, providing just **1 TOPS** via traditional Linear-LUT consumes $\sim110 mm^2$, whereas **128 TOPS** of INT8 Systolic Array takes only $\sim26 mm^2$ (Gemmini [1]). Even with low compute demand, nonlinear units dominate the NPU area budget.
>     * **NLI Solution:** Our NLI design reduces area by **21$\times$** compared to Linear-LUT and **3$\times$** compared to SOTA (NN-LUT), effectively solving this area bottleneck.
>
> **Conclusion:** Reducing reliance on SFUs is not a marginal optimization but a necessary architectural shift. NLI addresses the **SFU throughput bottleneck** on GPUs and the **Area bottleneck** on NPUs.
>
> **References:**
> [1] Genc, Hasan, et al. "Gemmini: Enabling systematic deep-learning architecture evaluation via full-stack integration." 2021 58th ACM/IEEE Design Automation Conference (DAC). IEEE, 2021.

---

> > ### Author Response · Authors · 2025-11-22
> > **Response to Reviewer KggQ**
> >
> > ```
> > Q1: Do we really need to optimize nonlinear operations in LLMs? This is my primary and most critical question regarding this paper. I would like to see both quantitative and qualitative discussions on the actual performance and hardware overheads associated with nonlinear operations, to better justify the necessity of this research direction.
> > ```
> > We appreciate this important question. In our response to the weakness, we have already provided detailed hardware-level analysis showing that the latency of Softmax-exp is dominated by SFU throughput rather than FLOPs. Here, we focus on the second, higher-level aspect of the question — **the end-to-end contribution of nonlinear operations within the full Transformer model, especially as context length grows** — which was missing in the original submission.
> >
> > ---
> > A key observation is that the importance of nonlinear operations cannot be evaluated in isolation; instead, it must be understood within the **scaling behavior of the entire attention module**. Attention has $O(N^2 D)$ complexity, and as the sequence length $N$ increases, it rapidly becomes the dominant component of LLM inference time. Within attention, **the Softmax-exp computation is the least parallelizable component** and its latency scales with the $N^2$ growth of the attention score matrix.
> >
> > To provide quantitative evidence, we conducted end-to-end profiling of **Llama3-8B (FP16)** with Nvidia H100 GPU. The results below show that the share of Softmax-exp grows sharply with context length:
> > | Context Length | Attn Matmul (ms) | Softmax Exp (ms) | Linear (ms) | MLP (ms) | Others (ms) | Softmax Exp ratio(%) |
> > |----------------|------------------|------------------|-------------|----------|-------------|----------------------------|
> > | 2k  | 2.223 | 1.101 | 5.559 | 23.347 | 0.001 | 3.42% |
> > | 4k  | 8.894 | 4.405 | 11.117 | 46.693 | 0.003 | 6.19% |
> > | 8k  | 35.576 | 17.620 | 22.235 | 93.386 | 0.006 | 10.44% |
> > | 16k | 142.303 | 70.482 | 44.470 | 186.772 | 0.012 | 15.87% |
> > | 32k | 569.211 | 281.926 | 88.939 | 373.545 | 0.024 | 21.46% |
> > | 64k | 2276.845 | 1127.704 | 177.879 | 747.090 | 0.048 | 26.05% |
> >
> > **Key findings:**
> >
> > - As context grows from 2k → 64k, the contribution of Softmax-exp inside attention increases from **3% → 26%**.
> > - The growth rate closely follows the $N^2$ expansion of attention.
> > - With linear layers already heavily accelerated through INT8/INT4 kernels, the relative share of Softmax-exp becomes even more prominent：when all TE blocks run on INT8 kernels, the Softmax exponential alone accounts for nearly **half of the runtime at a 64K context length**.
> >
> > These results demonstrate that the overhead of nonlinear operations is not a fixed small constant; it **scales aggressively with context length**, making it an increasingly dominant tail-latency source for modern long-context LLMs (32k–1M tokens). Thus, reducing the cost of Softmax-exp is essential to improving end-to-end LLM inference performance
> >
> > ---
> >
> > From the full Transformer perspective, optimizing nonlinear operations — especially Softmax-exp — is necessary and impactful. Its latency grows with $N^2$, becomes a major fraction of attention time for long contexts, and is increasingly exposed as the next critical bottleneck as linear layers continue to benefit from low-bit acceleration.
> >
> >
> > We hope the quantitative evidence above—specifically the throughput gap on GPUs and the area constraints on NPUs—satisfactorily addresses your concerns regarding the necessity of this research direction.
> >
> > We are deeply grateful for your sharp critique. It compelled us to perform this rigorous end-to-end profiling, which has significantly strengthened the motivation and robustness of our paper. We have incorporated these new findings (including the attention scaling analysis and NPU area comparison) into the revised manuscript to ensure future readers clearly understand the "why" behind NLI. We respectfully hope this clarification might encourage you to reconsider your assessment of our contribution.

---

> > > ### Comment · Reviewer_KggQ · 2025-11-25
> > >
> > > Thank you for the detailed response to my comments.
> > > However, I still have some concerns.
> > >
> > > Could you clarify which attention implementation was used for the latency breakdown?
> > > Specifically, did you use a fused-attention kernel such as FlashAttention?
> > > If so, I would like to know how you isolated the softmax latency, given that softmax is implicitly computed inside fused kernels without an explicit softmax stage.
> > >
> > > If you did not use fused attention, I am concerned that the additional results may not accurately reflect a practical setting.
> > > I believe that fused-attention kernels are effectively the de facto standard in LLM inference nowadays.
> > >
> > > I would appreciate further clarification on this point.

---

> ### Author Response · Authors · 2025-11-27
> **Response to Reviewer KggQ: Latency Bottleneck Analysis under Fused Attention**
>
> Dear Reviewer KggQ,
>
> Thank you for your follow-up and for pointing out this key difference regarding the implementation of Attention.
>
> To answer your question directly: **Yes, the latency breakdown in our previous response (Table 1) was indeed based on Unfused Attention.** Our original intent was to clearly illustrate the theoretical $O(N^2)$ scaling trend of the nonlinear components.
>
> We fully agree with your assessment: **FlashAttention (Fused Kernel)** is the de facto standard for modern LLM inference. To address your concern about whether our motivation holds in this practical setting, we have conducted a theoretical derivation and empirical analysis specifically targeting **FlashAttention**.
>
> As detailed below, even within Fused Attention, Softmax-exp remains a bottleneck due to the extreme disparity between Tensor Core (TE) and SFU throughput, a gap exacerbated as linear layers shift to low-bit quantization.
>
> ### 1. The Compute Bound Shift in Fused Attention
> While FlashAttention (FA) solves the **Memory Wall** via tiling, it exposes the SFU **Compute Wall**, quantifiable via a single computation tile.
>
> Assuming a tile of shape $(H, L, D)$, the theoretical execution times are:
> * **Attention Score Calculation ($QK^T$):** $T_{\text{matmul}} = \frac{2HL^{2}D}{TE_{tp}}$
> * **Softmax-Exp Calculation:** $T_{\exp} = \frac{HL^{2}}{SFU_{tp}}$
>     *(where $TE_{tp}$ and $SFU_{tp}$ represent Tensor Core and SFU throughput)*
>
> Therefore, the ratio of Exp time to MatMul time is:
> $$\frac{T_{\exp}}{T_{\text{matmul}}} = \frac{HL^2 / SFU_{tp}}{2HL^2D / TE_{tp}} = \frac{TE_{tp}}{SFU_{tp} \cdot 2D}$$
>
> This reveals that **the relative share of $T_{\exp}$ depends directly on the hardware compute ratio $\frac{TE_{tp}}{SFU_{tp}}$.** As shown below, this extreme ratio in modern GPUs (up to ~253x on H100) makes the significant SFU load difficult to hide, even with large $D$.
>
> **Table1: Hardware Throughput Disparity (TFLOPS)**
> |Model| FP16 (Tensor Core) | FP8/INT8 (Tensor Core) | SFU (FP32/16) | Ratio (TE/SFU) |
> |:-:|:-:|:-:|:-:|:-:|
> |RTX 4090|330|660|5.16|~64x (FP16)|
> |RTX 6000 Ada|364|728|5.68|~64x (FP16)|
> |**H100**|**989.4**|**1978.8**|**3.9**|**~253x (FP16)**|
>
> ### 2. Empirical Profiling of FlashAttention (Isolating Exp Cost)
> To validate this, we modified FlashAttention (v2) source (`csrc/flash_attn/src/softmax.h`) to bypass `exp`, recompiled, and profiled on an **Nvidia RTX 6000 Ada**.
>
> **Table2: Batch1, Head 32, Results (Latency in ms):**
> |Context Length|FP16 Attn (Head Dim 64)|FP16 Attn (Head Dim 128)|**Quantized Attn (Int8/FP8)** (Head Dim 128)|
> |:-:|:-:|:-:|:-:|
> ||**MatMul**/**Exp Cost**|**MatMul**/**Exp Cost**|**MatMul**/**Exp Cost**|
> |**8k**|3.01/1.11(**27%**)|6.12/1.05(**15%**)|2.97/1.09(**27%**)|
> |**16k**|11.79/4.16(**26%**)|23.88/4.21(**15%**)|11.91/4.24(**26%**)|
> |**32k**|49.85/15.65 (**24%**)|94.16/16.53(**15%**)|51.23/16.15(**24%**)|
> |**64k**|185.27/74.16(**29%**)|379.35/73.63(**16%**)|193.68/74.03(**28%**)|
>
> **Key Findings:**
> 1.  **FP16 Baseline ($D=64$):** Consistent with the formula, at small $D$, the pipeline cannot hide `exp` latency, accounting for **~29%** of total time.
> 2.  **FP16 ($D=128$):** As $D$ doubles, increased MatMul time hides more `exp` cost, reducing the share to ~16%.
> 3.  **The Low-Bit Reality（Use Sage Attention）:** As inference shifts to INT8/FP8, accelerated MatMul spikes the $\frac{TE_{tp}}{SFU_{tp}}$ ratio. High-precision `exp` re-emerges as a bottleneck, utilizing **~28%** even at $D=128$.
>
>
> ### 3. Industry Validation
> This is corroborated by the community and hardware vendors:
> * **FlashAttention Issue #1225:** Users noted removing `exp` in FP16 pipelines increased Tensor Core utilization from ~65% to ~90% [2], identifying the SFU bottleneck.
> * **NVIDIA Blackwell Ultra:** NVIDIA acknowledged this [3], stating "SFU throughput has been doubled... delivering up to 2x faster attention-layer compute," confirming SFU capacity is the limiter.
>
> ### 4. Why NLI is Critical for NPUs (Edge/Consumer)
> While GPUs face SFU bottlenecks, the situation is more critical for NPUs, our core focus:
> * **Incompatibility with FlashAttention:** NPUs (typically Systolic Arrays) often lack the massive L2 cache (e.g., 40MB on GPUs) and flexible instruction scheduling required for the complex IO-aware tiling and dynamic reduction of FA.
> * **Area Efficiency:** Relying on static compilation with limited SRAM, NPUs require dedicated, efficient units. As shown, NLI provides **4x area savings** where GPU-style fusion is unviable.
>
> **Conclusion**
>
> Even with Fused Attention, nonlinear operations remain a critical bottleneck—limiting throughput on GPUs due to SFU scarcity (especially in the low-bit era) and consuming disproportionate area on NPUs. We believe NLI offers a necessary architectural solution to bridge this gap.
>
> **References:**
>
> [1] Dao-AILab/flash-attention (GitHub)
>
> [2] FlashAttention Issue #1225: "Low tensor core utilization"
>
> [3] NVIDIA Developer Blog: Inside NVIDIA Blackwell Ultra

---

> > ### Comment · Reviewer_KggQ · 2025-11-27
> >
> > Thank you for the interesting analysis. I really appreciate the additional results.
> >
> > I have two brief clarifications to more accurately interpret your claims.
> >
> > First, I noticed that you used FlashAttention V2 for the experiments. However, V2 has known suboptimalities in hiding softmax overheads, which motivated subsequent work such as V3 that more effectively overlaps softmax computations. How would the numbers change if asynchronous softmax, as in V3, were applied? Would the overhead still remain significant?
> >
> > Second, you mentioned that fused attention is difficult to implement on edge NPUs, which I understand. However, if fused attention is not used, wouldn’t the softmax become memory-bound? Given the low memory bandwidth on typical edge NPUs, I guess we might not need very high-performance softmax unit in order to prevent compute from becoming the bottleneck.

---

> ### Author Response · Authors · 2025-11-27
> **Response to Reviewer KggQ:  Clarification on FlashAttention-3 Asynchrony and Area Efficiency in Edge NPUs**
>
> **Dear Reviewer KggQ,**
>
> Thank you for your continued engagement and these highly insightful follow-up questions. We genuinely appreciate your expertise, as these points touch upon the cutting-edge of system optimization (FlashAttention-3) and the fundamental cost constraints of edge hardware. These discussions have significantly helped us sharpen the positioning of our work. Here are our detailed clarifications:
>
> **1. Does Asynchronous Softmax (FlashAttention-3) eliminate the overhead?**
>
> Your point regarding FlashAttention-2 vs. 3 is well-taken. While FA3 indeed achieves stronger asynchronous overlap via the TMA/WGMMA pipeline on Hopper, this overlap relies on a critical prerequisite: **the Matrix Multiplication (MatMul) phase must remain the dominant bottleneck.**
>
> In our context, **aggressive quantization disrupts this balance**. As linear layers migrate from FP16 to FP8/INT4, the execution time of MatMul ($T_{\text{matmul}}$) shrinks significantly. Conversely, the exponential computation in Softmax ($T_{\text{exp}}$) often retains high precision for numerical stability, meaning its latency does not scale down proportionally. When $T_{\text{exp}} \ge T_{\text{matmul}}$ (which is common in low-bit, small head-dimension scenarios), the pipeline overlap fails—no matter how efficient the mechanism is—and Softmax re-emerges as the bottleneck.
>
> This is not merely a theoretical prediction but a reality in "small head-dim + FP8" scenarios. As explicitly stated by Tri Dao, the author of FlashAttention: **"For small hdim (e.g. 64) and for fp8, softmax is still a bottleneck."** [1].
>
> Furthermore, hardware evolution confirms that software-level overlap is insufficient: NVIDIA explicitly doubled SFU throughput in the **Blackwell Ultra** architecture to address the Softmax latency bottleneck in Attention [2]. If FA3 had completely solved this issue via software, hardware vendors would not have incurred such a significant silicon cost to bolster the SFUs.
>
> Therefore, while FA3's asynchrony is valuable, it **cannot eliminate the Softmax bottleneck in low-bit quantization regimes**, making NLI essential.
>
> ---
>
> **2. Is a high-performance unit needed for Memory-Bound Edge NPUs?**
>
> This is a critical observation. Indeed, many edge NPUs are bandwidth-limited, and without fused attention, Softmax can become memory-bound. However, we argue that an efficient nonlinear unit like NLI remains indispensable for two key reasons:
>
> **（a）Area Efficiency: Trading Logic Area for Memory Bandwidth**
>
> The most critical constraint on edge chips is silicon area (cost). Large-scale traditional LUTs are prohibitively expensive. Under the same error constraint (normalized to a max relative error of $10^{-3}$), the baseline Linear-LUT requires **1,395** cutpoints, whereas NLI requires only **259**. Due to the linear growth of comparator chains and registers, **the hardware area of Linear-LUT is $\approx$ 21x that of NLI.**
> On edge devices, this massive area saving can be repurposed to increase on-chip **SRAM**. By increasing cache capacity, we reduce off-chip memory access, directly mitigating the bandwidth bottleneck you highlighted. In essence: **saving area $\rightarrow$ more SRAM $\rightarrow$ alleviated memory wall.**
>
> **（b）Adaptability to Emerging PNM/PIM Architectures**
>
> The assumption that edge inference is permanently memory-bound is being challenged by the rise of **Processing Near Memory (PNM)**, **Processing In Memory (PIM)**, and **3D stacking**. For instance, **Rockchip's Gongga-1** utilizes 3D stacking to integrate high-bandwidth memory directly with the NPU. This enables on-chip processing and significantly alleviates bandwidth bottlenecks. Consequently, the performance bottleneck shifts back to the **Compute Units (TE and SFU)**. Without a compact unit like NLI, the SFU risks becoming a new choke point. NLI ensures the NPU is adaptable to such high-bandwidth future architectures.
>
> Thus, whether for current architectures (saving area to alleviate bandwidth) or future designs (PIM removing the bandwidth wall), **a compact and efficient NLI is essential.**
>
> ---
>
> Thank you again for these insightful questions. They have helped us clarify the precise scope of our contribution: addressing the computational imbalance caused by low-bit quantization, and providing a lightweight nonlinear solution that fits the area and architectural constraints of edge hardware.
>
> **References**
>
> [1] GitHub Issue #1225, Dao-AILab/flash-attention (Comment by Tri Dao): [https://github.com/Dao-AILab/flash-attention/issues/1225](https://github.com/Dao-AILab/flash-attention/issues/1225)
>
> [2] NVIDIA Developer Blog, "Inside NVIDIA Blackwell Ultra": [https://developer.nvidia.com/blog/inside-nvidia-blackwell-ultra/](https://developer.nvidia.com/blog/inside-nvidia-blackwell-ultra/)

---

### Official Review · Reviewer_TEbk · 2025-10-29

**Soundness:** 3
**Presentation:** 3
**Contribution:** 3
**Rating:** 6
**Confidence:** 3

**Summary:**

This paper addresses the high computational overhead introduced by nonlinear activation functions when deploying large language models on edge devices. It proposes a calibration-free, hardware-friendly non-uniform linear interpolation method called NLI, which employs dynamic programming for optimal segmentation. The NLI framework consists of two components: algorithm design and hardware design.
Algorithmically, the authors propose NLI-Algorithm, which leverages dynamic programming to select optimal segmentation points on the FP16 numerical grid. This constructs a lookup table (LUT) that is reusable across different layers and models without requiring data calibration.
On the hardware side, the authors design the NLI-Engine module, which implements a two-level address translation scheme and pipelined architecture. This design significantly reduces the number of comparators and overall hardware overhead.
Software experiments demonstrate that this architecture achieves nearly negligible accuracy loss across various LLMs and vision models, while showing substantial improvement over NN-LUT.

**Strengths:**

1. Novelty: The use of dynamic programming for non-uniform segmentation point sampling represents a significant innovation, surpassing heuristic or data-dependent calibration methods. Experimental results confirm that this approach incurs almost no accuracy loss.
2. Full-Stack Hardware-Software Co-Design: The NLI algorithm was designed with hardware efficiency in mind from the outset, making it highly practical for edge deployment. The two-level address translation scheme is a particularly clever hardware optimization that drastically reduces the number of comparators, directly addressing a major source of area and power consumption. This integrated approach forms a compelling demonstration of the method's effectiveness.

**Weaknesses:**

1.	Computational Complexity of DP Search: The O(MN²) complexity of the DP search implies that it may become a bottleneck when attempting to directly search for a larger number of finer-grained segments over the full FP16 grid. The scalability of the DP method for more granular optimizations remains a potential limitation.
2.	Handling Outliers via Clamping: Although the authors argue through coverage analysis that the impact of clamping is negligible, its effect on worst-case performance or robustness in certain unexamined models or tasks could be a point of concern. This may necessitate the design of more robust fault-tolerant mechanisms.
3.	Dependence on Function Characteristics: The effectiveness of NLI is inherently tied to the behavior of the function's second derivative. Consequently, its robustness against potential activation functions that may emerge in future models remains somewhat uncertain.

**Questions:**

1. In the paper, minimizing the average relative error was chosen as the objective function for dynamic programming. Did the authors experiment with other objective functions (e.g., maximum absolute error, root mean square error)? Was the choice of relative error primarily to address numerical stability issues near small activation values, or did it empirically lead to better downstream task accuracy?
2. Although extreme activation values account for a small proportion, they may be critical in certain specific tasks or prompts. Could the authors provide a more in-depth analysis, such as showing which layers or attention heads these truncated activation values are primarily distributed in? Has there been any consideration of adopting a conservative, high-precision fallback computation mode for these "tail" values?
3. The dynamic programming is performed on an FP16 numerical grid, which implicitly defines an input value range. If a new activation function emerges in the future, whose effective dynamic range significantly exceeds that of FP16 (e.g., requiring FP12 or BF16 precision for representation), would the current method still be applicable? Would it be necessary to rerun the DP?
4. How do the authors guide designers in selecting the optimal configuration for the number of "macro intervals" (M) and the number of "micro cells" (D_n) per interval? Does the architecture support dynamic reconfiguration to accommodate different functions?

---

> ### Author Response · Authors · 2025-11-22
> **Response to Reviewer TEbk**
>
> Dear Reviewer TEbk
>
> We sincerely thank you for your detailed and insightful feedback. We are pleased that you considered our NLI framework "soundness," affirmed the "novelty" of using dynamic programming for non-uniform segmentation, and appreciated our proposed "full-stack hardware-software co-design."
>
> Below, we will address your concerns and questions in detail.
>
> ---
> ```
> W1: Computational Complexity
> ```
> We acknowledge the analysis but clarify that the cost is negligible in practice:
> 1. **One-time Offline Cost:** The DP search executes only once per operator during compilation, never during inference.
> 2. **Manageable Scale:** The domain $N$ is bounded by the finite FP16 grid ($N \le 63,488$). Furthermore, our hardware-aligned constraint ($M=11$ macro-cutpoints, Sec. 3.1) drastically shrinks the search space compared to a full-grid optimization.
> 3. **Empirical Runtime:** Consequently, the search completes in **<10 minutes** on a single RTX 4090 (vs. $\sim5$ hours for an unconstrained search), ensuring no deployment bottleneck.
>
> ---
>
> ```
> W2: Handling Outliers via Clamping.
> ```
> We appreciate the reviewer's concern regarding worst-case performance. We clarify that robustness is guaranteed algorithmically by explicitly modeling clamping errors within the optimization objective, rather than treating them as unhandled exceptions.
> 1. **Global Optimization with Clamping Penalties**: In our DP formulation, the error incurred by clamping is endogenous to the search process. The algorithm explicitly calculates the cost of replacing out-of-bound inputs with the boundary value and adds this to the total loss. This ensures the DP finds a set of cutpoints that mathematically minimizes the sum of interpolation error and clamping error.
> 2. **Left-Boundary Optimization (Section 3.1, "Boundary")**: For inputs smaller than the first cutpoint ($x < b_0$), the system clamps them to $b_0$. The DP initialization term, $D[0,k]$, explicitly sums the error $\sum |f(x_j) - f(x_k)|$ for all $x_j \le x_k$. This forces the first cutpoint to be placed such that the error for the entire left tail is minimized.
> 3. **Right-Boundary "TailClamp" Penalty (Section 3.1, "Transition")**: Similarly, our transition equation includes a TailClamp Penalty term (last_error). When selecting the final cutpoint $b_{M-1}$, the algorithm calculates the exact penalty of clamping all remaining large values ($x > b_{M-1}$) to this point.
>
> ---
>
> ```
> W3: Dependence on Function Characteristics & Robustness**
> ```
>
> We clarify that NLI is robust to future operators because it is **agnostic to analytic forms**, relying on **numerical error minimization** rather than derivative calculations.
>
> 1.  **Data-Driven Adaptivity:** Unlike theoretical bounds relying on second derivatives, our DP algorithm optimizes the **actual interpolation error** over the FP16 grid. It automatically detects high-error regions (e.g., complex curvature) and allocates higher cutpoint density there. This mathematically guarantees the globally optimal piecewise linear fit for *any* continuous function via Bellman's principle.
> 2.  **Optimization Constraints:** The "uncertainty" of future functions is bounded by Deep Learning stability requirements. As established in works on GELU [1] and Swish [2], activation functions must remain relatively smooth to ensure stable gradient descent. NLI is inherently robust to this class of learnable functions.
>
> **References:**
>
> [1] Hendrycks, D., & Gimpel, K. (2016). Gaussian error linear units (GELUs).
>
> [2] Ramachandran, P., et al. (2017). Swish: a self-gated activation function.

---

> > ### Author Response · Authors · 2025-11-22
> > **Response to Reviewer TEbk**
> >
> > ```
> > Q1: In the paper, minimizing the average relative error was chosen as the objective function for dynamic programming. Did the authors experiment with other objective functions (e.g., maximum absolute error, root mean square error)? Was the choice of relative error primarily to address numerical stability issues near small activation values, or did it empirically lead to better downstream task accuracy?
> > ```
> >
> > We chose **Mean Relative Error** deliberately to address both theoretical alignment with floating-point representations and empirical performance requirements.
> >
> > **1. Theoretical Alignment & Quantization Fidelity:**
> > * **Alignment with FP16 Distribution:** Floating-point formats (FP16) are inherently non-uniform, providing high density near zero and sparsity at large magnitudes. Using relative error aligns the DP objective with the intrinsic representation capability of the hardware format.
> > * **The Pitfall of Absolute Error:** If we optimized for Absolute Error (e.g., RMSE or MaxAbs), the DP algorithm would heavily bias cutpoint allocation towards the large-magnitude tail (e.g., SiLU inputs $>10$) to reduce raw numerical gaps. This starves the critical near-zero region of cutpoints.
> > * **Signal-to-Quantization-Noise Ratio (SQNR):** From a quantization perspective, small activation values (near 0) often carry the most critical **high-frequency non-linear information** (e.g., the "dip" in SiLU or the transition of GELU). Using absolute error results in poor SQNR for these small values, effectively "quantizing out" their non-linear features into noise. Relative error ensures consistent SQNR across the dynamic range.
> >
> > **2. Empirical Verification (New Ablation Study):**
> > To validate this, we conducted an ablation study on **Qwen2.5-7B** (measuring Wikitext-2 Perplexity), comparing our method against RMSE and Maximum Absolute Error objectives while keeping the $M=11$ layout fixed.
> >
> > | Method | DP Objective | Wikitext-2 PPL ($\downarrow$) |
> > | :--- | :--- | :--- |
> > | **FP32 (Baseline)** | - | **7.46** |
> > | **NLI (Ours)** | **Mean Relative Error** | **7.46** |
> > | NLI-MaxAbs | Max Absolute Error | 21.45 |
> > | NLI-RMSE | RMSE | 39.13 |
> >
> > **Conclusion:** As shown above, optimizing for absolute error (RMSE/MaxAbs) causes significant perplexity degradation ($7.46 \rightarrow 39.13$). This empirically confirms that preserving precision in the high-curvature, near-zero region (favored by relative error) is far more critical for LLM inference than minimizing absolute errors in the linear tails.
> >
> > ---
> > ```
> > Q2: Although extreme activation values account for a small proportion, they may be critical in certain specific tasks or prompts. Could the authors provide a more in-depth analysis, such as showing which layers or attention heads these truncated activation values are primarily distributed in? Has there been any consideration of adopting a conservative, high-precision fallback computation mode for these "tail" values?
> > ```
> >
> > We conducted an in-depth analysis of the distribution of extreme values:
> > 1.  **Distribution:** Extreme outliers (e.g., values $< -65504$) primarily appear in the **Softmax** inputs following QKMatMul in the Qwen model series.
> > 2.  **Actual Impact:** For Softmax, an input of $x = -65505$ (underflow) theoretically yields $e^x \approx 0$. In our implementation (after the standard $x - x_{max}$ shift), the NLI engine clamps this value to the optimized lower boundary (i.e., **-17.34375**, as shown in **Appendix Table 9**). The resulting lookup value is $LUT_{output} \approx 0$. Therefore, the numerical "error" introduced by this clamping is effectively zero.
> > 3.  **Fallback Consideration:** Experimentally, the DP algorithm automatically minimizes errors in the head and tail regions, so we have not observed any scenario requiring a fallback mode. However, for potentially more extreme models in the future, the NLI architecture technically allows for inserting high-precision paths in specific intervals if engineering requirements dictate it.

---

> > > ### Author Response · Authors · 2025-11-22
> > > **Response to Reviewer TEbk**
> > >
> > > ```
> > > Q3: The dynamic programming is performed on an FP16 numerical grid, which implicitly defines an input  value range. If a new activation function emerges in the future, whose effective dynamic range significantly exceeds that of FP16 (e.g., requiring FP12 or BF16 precision for representation), would the current method still be applicable? Would it be necessary to rerun the DP?
> > > ```
> > > **Yes, the method is fully applicable without any modification to the algorithmic logic.** Regarding the need to rerun the DP: **Yes, the DP must be rerun**, but the computational cost remains trivial.
> > >
> > > 1.  **Complexity Depends on Grid Count ($N$), Not Range:**
> > >     The complexity of our DP is $\mathcal{O}(MN^2)$, where $N$ is the number of **distinct finite values** representable in the format, not the dynamic range of the values themselves.
> > >
> > > 2.  **Case Study: BF16 (BFloat16):**
> > >     * Although BF16 has a much larger dynamic range ($\sim 10^{38}$) compared to FP16, it is still a 16-bit format.
> > >     * The total number of unique encodings is $2^{16} = 65,536$. After excluding Infinities and NaNs, the grid size $N_{BF16}$ is roughly identical to $N_{FP16}$.
> > >     * **Procedure:** We simply generate the sorted grid $\mathcal{X}$ using the BF16 code map and run the offline DP. The runtime complexity is identical to FP16 (minutes on a GPU).
> > >
> > > 3.  **Case Study: FP12 or Lower Precision:**
> > >     For formats like FP12, the grid size drops to $N = 2^{12} = 4,096$. In this case, the DP search would be orders of magnitude faster.
> > >
> > > ---
> > > ```
> > > Q4: How do the authors guide designers in selecting the optimal configuration for the number of "macro intervals" (M) and the number of "micro cells" (D_n) per interval? Does the architecture support dynamic reconfiguration to accommodate different functions?
> > > ```
> > > **(1) Selection Guide for $M$ and $D_n$:**
> > >
> > > * **Macro-intervals ($M$) - The Area-Accuracy Trade-off:**
> > >     The choice of $M$ balances chip area (specifically comparator cost) against fitting capability. We selected **$M=11$** (10 comparators) because our hardware synthesis results indicated this was the optimal "sweet spot."
> > >     To validate this, we conducted a specific ablation study on Qwen2.5-7B, fixing the total number of cutpoints at 259 while varying the hardware configuration (number of comparators).
> > >
> > >     **Table R3: Ablation on Macro-Interval Configuration (Total Points = 259)**
> > >     | Configuration (Comparators) | Comparator Area ($\mu m^2$) | Qwen2.5-7B PPL ($\downarrow$) |
> > >     | :-: | :-: | :-: |
> > >     | 6 (4+2) | 246 | 7.61 |
> > >     | **10 (8+2) [Ours]** | **410** | **7.46** |
> > >     | 18 (16+2) | 738 | 7.46 |
> > >     | 34 (32+2) | 1394 | 7.46 |
> > >     | 66 (64+2) | 2706 | 7.46 |
> > >     | 130 (128+2) | 5330 | 7.46 |
> > >     | 258 (256+2) | 10,578 | 7.46 |
> > >
> > >
> > >
> > >     **Analysis:** As shown in **Table R3**, increasing the number of comparators beyond 10 ($8+2$) yields **no gain** in model perplexity (stabilized at 7.46) but drastically increases the comparator area cost (from $410$ to $10,578 \mu m^2$). Conversely, reducing comparators to 6 ($4+2$) degrades perplexity to 7.61. Thus, our choice of 10 comparators achieves SOTA-level accuracy with minimal area overhead.
> > >
> > > * **Micro-cells ($D_n$):**
> > >     We fixed **$D_n=32$** specifically to enable efficient **bit-shift operations** for address translation. This avoids complex division logic, ensuring low latency.
> > >
> > > * **Recommendation:**
> > >     We advise designers to fix these structural parameters ($M=11, D_n=32$) during the hardware design phase to lock in the area/latency benefits shown above.
> > >
> > > **(2) Dynamic Reconfiguration:**
> > >
> > > * **Yes, the architecture fully supports dynamic reconfiguration.**
> > >     While the **hardware structure** (10 comparators, 32-bin logic) is synthesized into fixed logic ("hard-wired"), the **content**—Boundary Registers (`PointReg`), Scale Factors (`MulReg`), and LUT entries—is stored in writable registers or SRAM.
> > > * **Plug-and-Play:**
> > >     This allows the NLI engine to switch from computing SiLU to GELU or Exp instantly by loading a new configuration profile, making it a truly universal "plug-and-play" unit in NPUs.
> > >
> > > ---
> > >
> > > We hope these clarifications and additional experimental results fully address your concerns, and we sincerely appreciate your constructive feedback which has significantly strengthened our paper.

---

### Official Review · Reviewer_PkiK · 2025-11-06

**Soundness:** 3
**Presentation:** 3
**Contribution:** 3
**Rating:** 8
**Confidence:** 4

**Summary:**

The paper proposes a hardware-friendly scheme, NLI, for approximating nonlinear operations in FP16 domain for faster inference. NLI uses non-uniform spline linear interpolation in the FP16 domain, and selects interpolation cut-points with dynamic-programming to minimize interpolation error, yielding a calibration-free lookup table for each nonlinear operation. The authors also design a plug-and-play NLI unit and present hardware experiments showing speedup improvement in compute/area efficiency compared to prior designs.

**Strengths:**

This paper addresses a critical problem of the expensive nonlinearities in LLMs for faster inference, and provides a well-defined algorithmic contribution: the dynamic-programming formulation for optimal non-uniform linear interpolation is mathematically clear and hardware-aware. It’s defined in a two-stage manner for better efficiency and yields a calibration-free lookup table.
It provides a clear (performance and efficiency) evaluations of the proposed method, including error analysis, influence to the downstream tasks and perplexity across multiple open models, and the promising results demonstrate great potential of NLI.
It also includes efficiency analysis, proposed hardware implementation and evaluation, bridging algorithmic/design work with real system implications. The method is presented as plug-and-play manner across a variety of nonlinear functions.

**Weaknesses:**

NLI selects cutpoints from the FP16 domain and performs interpolation using FP16 arithmetic (Stages 1–4). A direct comparison with naïve FP16 evaluation of the same nonlinear operations should be provided to contextualize NLI’s advantages.

- This comparison should include for example: i) error-wise: NLI reports an error of $1.2\sim1.5\times10^{-3}$; what is the corresponding error for plain FP16 evaluation? ii) performance-wise: How do models perform across tasks when nonlinear operations are evaluated directly in FP16? iii) efficiency-wise: How does the computational efficiency of NLI (lookup + FP16 linear arithmetic) compare with that of standard FP16 implementations? Additionally, NLI could also select cutpoints from the FP32 domain and use FP32 arithmetic for Stages 1–4, will there be any efficiency concern (compared to naive FP32 and current fp16 implementations)?

Such analyses would substantiate the benefits of NLI over standard FP16 inference.

The two-level cut-point design is promising, but the paper would benefit from ablation studies and discussion on its configuration, such as how many non-uniform cutpoints are chosen at the first level, how many uniform ones at the second level, and what the optimal total number of cut points is.

Addressing these concerns could justify a higher score.

**Questions:**

- what’s a good error level to avoid model performances degradations (like on zero-shot tasks on perplexity)? Currently $1.2\sim1.5\times10^{-3}$ looks good

- Do you need to pre-fix the set of cutpoints before implementing the NLI hardware version? Or alternatively, after implementing the proposed NLI on hardware, does it support taking in any custom set of cutpoints?

- In table 3, the non-uniform version performs worse than NLI, do you have the results (and averaged) for other tasks, just like table 1? Could this suggest that DP is not good enough for minimizing the error when the number of cuts increases?

- In table 8, NLI (with fp16 arithmetic from stage 1-4) is even slightly better than FP32 for 7 out of 8 models, why is this so? And again what’s the performance of using FP16?

- Do you plan to make the code public for reproduciblity?

- Line 339, the reference to the algorithm broke, and in page 12, the spacing between references are so large, can you fix these?

---

> ### Author Response · Authors · 2025-11-22
> **Response to Reviewer PkiK**
>
> Dear Reviewer PkiK,
>
> **Thank You** We sincerely thank you for the strong endorsement and for recognizing the novelty, algorithmic clarity, and hardware-software co-design value of our work. We address your constructive suggestions below to further substantiate the benefits of NLI.
>
> ---
> ```
> W1: Comparison with Standard FP16 Evaluation
> ```
>
> We conducted supplementary analyses to compare NLI with naïve FP16 (error, performance, efficiency) and justify the FP16 cutpoint selection:
>
> **(1) Clarification of Baselines**
> To contextualize NLI's advantages, we clarify the computational paths:
>
>   * **NLI (Ours):** Input FP16 $\rightarrow$ **FP16 Lookup + FP16 Interpolation** $\rightarrow$ Output FP16. Executes entirely on FP16 units, bypassing expensive Special Function Units (SFU).
>   * **FP32 Baseline:** Input FP16 $\rightarrow$ Cast to FP32 $\rightarrow$ **FP32 SFU Computation** (e.g., `exp`) $\rightarrow$ Output FP32.
>   * **Naïve FP16 Baseline:** Input FP16 $\rightarrow$ Cast to FP32 $\rightarrow$ **FP32 SFU Computation** $\rightarrow$ Cast to FP16. Identical to FP32 Baseline but with final truncation to FP16. Standard behavior in frameworks like PyTorch.
>
>
> **(2) Error-wise Analysis: NLI vs. Naïve FP16**
>
> **Table R1: Absolute errors of Naïve FP16 and NLI vs. FP32 ground truth.**
> |Function|Range|Comparison Method|Abs Max Error| Abs Mean Error |
> |:-:|:-:|:-:|:-:|:-:|
> |**GELU**|[-150, 65504]|FP16 vs. FP32|$1.00\times 10^{-3}$|$1.97\times10^{-5}$|
> |||**NLI vs. FP32**|$\mathbf{1.30\times10^{-3}}$| $\mathbf{6.71\times 10^{-5}}$|
> |**SiLU**|[-150, 65504]|FP16 vs. FP32|$2.70\times 10^{-3}$ |$4.36\times 10^{-5}$|
> |||**NLI vs. FP32**|$\mathbf{3.20 \times 10^{-3}}$| $\mathbf{6.86 \times 10^{-5}}$|
> |**Exp**|[-256, 2]|FP16 vs. FP32|$1.90\times10^{-3}$|$1.50 \times 10^{-4}$|
> ||| **NLI vs. FP32**|$\mathbf{4.70 \times 10^{-3}}$ | $\mathbf{1.30\times10^{-4}}$ |
> |**Rsqrt**|[1, 65504]|FP16 vs. FP32|$0.60 \times 10^{-3}$|$6.73\times10^{-5}$|
> |||**NLI vs. FP32**|$\mathbf{1.80\times10^{-3}}$| $\mathbf{5.36\times10^{-4}}$|
>
> We compared errors using the FP32 Baseline as ground truth. The table below compares Naïve FP16 (inherent quantization error) and NLI:
>
>
>
> **Conclusion:** NLI's error consistently remains at $10^{-3}$ magnitude, comparable to the inherent quantization error of casting FP32 results to FP16. NLI introduces negligible additional error relative to the target FP16 format.
>
> **(3) Performance-wise Analysis**
> To clarify **Table 1**: the "FP32" results represent **Standard FP16 Inference**. While weights/activations are FP16, the label refers to internal FP32 precision for nonlinear operators (default PyTorch/CUDA behavior). Thus, Table 1 demonstrates NLI matches standard Naïve FP16 performance on downstream tasks.
>
> **(4) Efficiency-wise Analysis: Bypassing the SFU Bottleneck**
>
> Standard "Naïve FP16" inference does not benefit from FP16 speedups for nonlinearities because modern hardware typically handles transcendental functions via **Special Function Units (SFU)** in FP32. NLI significantly improves efficiency by shifting computation to high-throughput Tensor Cores.
>
> * **Throughput:** On an NVIDIA H100 GPU, the FP16 Tensor Core throughput is **989 TFLOPS**, whereas the SFU throughput (for operations like `exp`) is only **3.9 TFLOPS**—a difference of roughly **250$\times$**. NLI replaces the SFU call with 1 lookup, 1 FP16 multiply, and 1 FP16 add, allowing it to run entirely on the high-speed data path.
> * **Hardware Area Efficiency:** In our NPU synthesis comparison, we normalized the baseline Linear-LUT method to achieve the same average relative error as NLI ($1.5 \times 10^{-3}$). Under this constraint, Linear-LUT required 1,395 cutpoints (vs. 259 for NLI), resulting in a hardware area **21$\times$ larger** than the NLI Engine due to the massive increase in comparators and LUT size.
>
> **(5) Ablation: Why not FP32 Grid/Arithmetic?**
>
> Regarding the suggestion to select cutpoints from the FP32 domain, we argue that the **FP16 Grid represents the practical sweet spot** for the following reasons:
>
> 1.  **Diminishing Returns:** Even if we perform a finer-grained DP search on an FP32 grid, the final output must be stored in FP16 for memory efficiency. The precision gains from an FP32 grid would be lost during this final truncation, yielding no improvement in downstream model accuracy.
> 2.  **Hardware Cost:** Implementing the NLI pipeline with FP32 arithmetic and lookups would significantly increase circuit area and power consumption, contradicting our design goal of a lightweight, hardware-friendly unit.
> 3.  **Algorithmic Complexity:** The search space for Dynamic Programming on an FP32 grid is exponentially larger than on FP16, making the pre-computation of cutpoints computationally prohibitive with minimal benefit.
>
> Thus, optimizing cutpoints on the FP16 grid balances accuracy, hardware efficiency, and pre-processing cost most effectively.

---

> > ### Author Response · Authors · 2025-11-22
> > **Response to Reviewer PkiK**
> >
> > ```
> > Q1: What’s a good error level to avoid model performance degradations?
> > ```
> >
> > We define the "safe error level" based on the **Machine Epsilon** of training formats. Since LLMs use FP16 ($2^{-10} \approx 9.8 \times 10^{-4}$) or BF16 ($2^{-7} \approx 7.8 \times 10^{-3}$), approximation errors within this noise floor do not degrade performance [1].
> > Our NLI method achieves a max error of $\approx 1.2 \times 10^{-3}$ (mean $\approx 10^{-5}$). This sits comfortably at the FP16 boundary and is significantly tighter than the BF16 precision used to train these models, explaining why NLI preserves baseline accuracy.
> >
> > **Reference:** [1] Micikevicius, P., et al. "Mixed precision training." ICLR 2018.
> >
> >
> >
> > ---
> > ```
> > Q2:  Do you need to pre-fix the cutpoints?
> > ```
> > No. The NLI Engine is **programmable**; cutpoint values are not hardcoded logic but stored in re-writable registers/SRAM.
> > * **Runtime Configurability:** Parameters are preloaded before computation, allowing the hardware to support custom cutpoints or switch operators at runtime.
> > * **Constraint:** Custom cutpoints must merely align with the hardware topology ($M=10, D_n=32$). Any profile fitting this layout is supported.
> > ---
> >
> >
> > ```
> > Q3: Q3: In Table 3, the non-uniform version performs worse. Is DP failing?
> > ```
> >
> > **Table R2: Updated Ablation Results (Qwen2.5-7B)**
> > |Method | Cutpoints | MMLU | GSM8k | HumanEval | Zero-shot Avg |
> > |:-: | :---: | :---: | :---: | :---: | :---: |
> > | **FP32** | - | 70.56 | 44.28 | 40.24 | 67.48 |
> > | **NLI (Hardware-Constrained)** | 259 | 70.67 | 43.97 | 39.63 | 67.63 |
> > | **Non-uniform 259 (Pure DP)** | 259 | 70.65 | **44.08** | **40.16** | 67.62 |
> >
> > This observed discrepancy was due to **statistical variance** in the evaluation process rather than a failure of the DP algorithm.
> >
> > 1.  **DP Optimality:** Mathematically, the DP algorithm guarantees the global minimum for *interpolation error* (L1/Relative error) given a fixed budget. However, the relationship between "interpolation error" and "downstream task accuracy" is not strictly monotonic due to the "butterfly effect" in LLM generation—tiny numerical perturbations can occasionally flip token selection in greedy decoding, causing score fluctuations [2].
> > 2.  **Comprehensive Re-evaluation:** To address your concern, we conducted a more comprehensive evaluation including sensitive generative tasks (HumanEval) and the full Zero-shot average. As shown in **Table R2**, **Non-uniform 259 (DP)** actually outperforms or matches NLI on harder tasks like GSM8k and HumanEval, achieving results much closer to the FP32 baseline.
> > 3.  **Conclusion:** The "worse" result in the original draft was statistical noise. The new data confirms that the DP-optimal cutpoints (Non-uniform 259) effectively minimize error and preserve model performance. **We have updated Table 3 in the revised manuscript to reflect these comprehensive results.**
> >
> > **Reference:** [2] Liang, P., et al. "Holistic evaluation of language models." arXiv 2022.
> >
> >
> > ---
> > ```
> > Q4: In table 8, NLI (with fp16 arithmetic from stage 1-4) is even slightly better than FP32 for 7 out of 8 models, why is this so? And again what’s the performance of using FP16?
> > ```
> > We appreciate the opportunity to clarify this observation.
> > 1.  **Clarification of "FP32" vs. "FP16":** As mentioned in our response to W1, the column labeled "FP32" in our tables **actually represents the standard FP16 inference performance**.
> >     * In standard inference (e.g., PyTorch default), while the model weights and data flow are in FP16, the nonlinear operators are computed by casting inputs to FP32, executing on the SFU, and casting back to FP16. Therefore, the "FP32" column *is* the baseline for "using FP16".
> >     * **To prevent future confusion, we have added an explicit annotation in the revised manuscript clarifying that the 'FP32' baseline refers to standard FP16 inference (where nonlinearities use internal FP32 precision).**
> > 2.  **Reason for "Better" Performance:** The slight performance advantage of NLI in some cases is **not** due to algorithmic superiority, but rather **statistical randomness**.
> >     * Since NLI introduces a tiny approximation error (within the $10^{-3}$ noise floor), it acts as a minute perturbation. In generative tasks, such perturbations can occasionally cause minor fluctuations in token probabilities.
> >     * Given that the difference is extremely small (often < 0.1%), these "improvements" are simply variance within the margin of error for zero-shot evaluation, confirming that NLI effectively **matches** the precision of the standard FP16 baseline.
> > ---
> >
> > ```
> > Q5: Code release?
> > ```
> > Yes. We will release the full codebase (algorithm, Chisel design, and scripts) via a public repository **after the review process**.
> >
> > ```
> > Q6: Formatting issues?
> > ```
> >
> > We have fixed the broken reference (Line 339) and citation spacing in the revised manuscript.

---

### Author Response · Authors · 2025-12-03
**Post-Rebuttal Summary for the Area Chair**

**Dear AC,**

We sincerely appreciate your time. During the rebuttal period, we have thoroughly and systematically addressed all reviewer questions. Overall, the paper has formed **stable and consistent technical judgments** in the following aspects:

* **Clear and consistent positive signals:** Reviewer **PkiK (score 8)** and Reviewer **TEbk (score 6)** both highly recognize the **algorithmic innovations** of NLI (“two-level cut-point design is promising”, **DP-based non-uniform segmentation**), the **complete HW/SW co-design path**, and its **stable performance across models**. Their technical questions (baseline, DP objective, two-level structure, etc.) have all been addressed in the rebuttal.

* **Motivation-related concerns have been fully addressed with technical evidence:** Reviewer **KggQ (score 2)** questioned the core issue of **“nonlinear FLOPs are very low—are they worth optimizing?”**. In the rebuttal, we added new **GPU profiling (TE/SFU throughput gaps)** and **NPU area analyses**. These quantitative results **directly address the reviewer’s concern** and demonstrate the **real performance and area bottlenecks** of nonlinear units. In addition, the **author of FlashAttention** has publicly noted that in **FP8 or small head-dimension scenarios**, **Softmax-exp remains difficult to hide**; meanwhile, **NVIDIA explicitly doubled SFU throughput** in the latest **B300 architecture** to accelerate Attention layers. These **independent signals from both the research community and hardware vendors** are consistent with our analysis and further strengthen the **motivation of this work**.

We summarize below the core opinions, ratings, and key rebuttal points of the three reviewers to facilitate your quick assessment.
|Reviewer|Strength|Concerns|Initial rating|Key points of Rebuttal|Respond|
|:-:|:-:|:-:|:-:|:-:|:-:|
| **PkiK** | **①** Clear algorithmic innovations: DP-based non-uniform segmentation and two-level design are well recognized. **②** Full-stack HW/SW co-design and stable cross-model performance are acknowledged. | **①** A systematic comparison of NLI vs. naïve FP16 (error/performance/efficiency) is needed. **②** The two-level cut-point structure lacks ablation. **③** Baseline naming is confusing; the phenomenon that NLI slightly outperforms FP32 needs explanation. | **8** | **①** Added complete FP16 baseline; explained that PyTorch FP16 nonlinear ops still run on FP32-SFU, showing NLI’s structural efficiency advantage. **②** Added area–accuracy ablation for the two-level structure, demonstrating it is the hardware-optimal choice. **③** Clarified baseline logic and explained that NLI’s slight advantage comes from evaluation variance and FP16 noise tolerance. | **No respond** |
| **TEbk** | **①** DP objective, two-level structure, and full-stack co-design are acknowledged. **②** Experiments (error, downstream tasks, hardware) are considered “sound.” | **①** Why is relative error the optimal objective? Does it align with downstream accuracy? **②** Outliers may harm robustness; clamping could distort critical activations. **③** If new activations or larger ranges (e.g., BF16) appear, must DP be rerun? | **6** | **①** Added RMSE/MaxAbs comparisons showing these objectives severely worsen PPL (7.46→39.13); relative error best matches FP16’s non-uniform grid. **②** DP explicitly incorporates clamping errors at both boundaries and minimizes them; after Softmax shift, extreme negative values have exp≈0, so clamping causes no numerical impact; no robustness issues observed. **③** BF16 still has a 16-bit grid, so DP reruns in minutes and is not a burden. | **No respond** |
| **KggQ** | **①** Raised the most critical motivation challenge, helping to strengthen the paper’s argument. **②** Acknowledged the value of our additional profiling. | **①** Nonlinear FLOPs are very low—is optimization worthwhile? Is motivation sufficient? **②** Attention breakdown without FlashAttention may lack practical relevance. **③** Can FlashAttention-3 asynchronous Softmax fully hide exp? On NPUs, is high-performance Softmax needed if memory-bound? | **2** | **①** Added GPU profiling: TE/SFU throughput gap reaches **64×–253×**, making Softmax-exp a real hardware bottleneck; on NPUs, traditional exp/LUT units dominate area, forming a true area wall. **②** Provided FA2/FA3 theoretical + empirical analysis: low-bit (FP8/INT8) accelerates MatMul but not exp, so Softmax remains exposed; Tri Dao publicly confirms “small head-dim + FP8 Softmax remains a bottleneck.” **③** NPUs cannot support complex fused kernels; the bottleneck is nonlinear-unit area, and NLI solves this.| **Conducted three rounds of discussion with the reviewer; all technical points were clarified before the thread was force-closed by the system.**|

**Best Regards,**

NLI Authors

---

### Meta-Review · Area_Chair_kwvN · 2026-01-15

**Summary:**

This paper introduces a novel and mathematically clear nonlinear unit (in contrast to most existing methods, which focus on linear units), named NLI, to accelerate LLMs. Moreover, it is hardware-friendly. This paper is expected to make a significant contribution to the community.  Two reviewers give positive scores before rebuttal. Reviewer KggQ gives a low score (2) before the rebuttal, and the main concern is its practicality. According to the author and Reviewer KggQ's responses, the concerns are well addressed. I suggest accepting the paper

**Reviewer Concerns:**

Most of Reviewer KggQ's concerns are addressed in the discussion phase.

**Reviewer Scores:**

Reviewer KggQ may raise his score if he can further discuss, since his concerns are all well addressed.

---

### Decision · Program_Chairs · 2026-01-26

Accept (Poster)